# FLYORIEN: A BIO-INSPIRED MODEL FOR INCREMENTAL LEARNING OF OBJECT ORIENTATION

## ABSTRACT

Visual orientation detection helps navigation, especially without a reliable magnetic compass or GPS. Inspired by the neural mechanisms of the insect brain, particularly the mushroom body (MB) and the central complex (CX), we propose FlyOrien—a bio-inspired model for object orientation detection. The model mimics the MB for random feature extraction, sparse coding and associative learning, while the CX provides multi-clue sensory integration, enabling interpolation for finer orientation representation. FlyOrien's biologically plausible learning rule allows one-shot learning, reducing the need for large datasets and repeated training. We tested FlyOrien on a dataset containing images labeled with orientations, which introduce strong interferences because images of the same object have different labels. In this challenging context, FlyOrien achieves competitive performance compared to convolutional neural networks (CNNs), significantly reducing training time and computational resources. It also has the potential for real-world applications like robotics, where incremental learning is essential.

## 1 INTRODUCTION

In natural environments, various cues like sun direction, skylight polarization, wind direction, and landmarks help animals navigate (Heinze, 2017). Most of these cues are perceived visually. Even simple insects can use visual memory to remember the way home after traversing a route once, leveraging mechanisms partly explained by the mushroom body (MB) (Ardin et al., 2016); for a review, see Modi et al. (2020). Their lightweight neural circuits outperform typical artificial neural networks (ANNs) in remembering orientations. Inspired by this, we investigated these circuits to develop an architecture and learning rule for retrieving orientation memory from visual signals.

Assuming an observer always faces an object, with a reference direction which could be true north, there are three orientations: the angle the observer is facing $o$, the angle the object is facing $o'$, and their relative angle $o - o'$. Knowing any two allows computation of the third. If $o$ and $o - o'$ are known, it is an object-orienting problem; if $o'$ and $o - o'$ are known, it is an observer-orienting problem. For simplification, in the object-orienting problem, $o$ is set to 0, and in the observer-orienting problem, $o'$ is set to 0. Hence, in our dataset, there is only one number as a label for each sample, and the two problems are not explicitly distinguished. By discretizing the range from $0°$ to $360°$ to multiple discrete values, the object orientation detection task can be set as a multi-class classification problem.

There have been many models for finding objects' orientation in the image plane but not horizontally on the ground, such as PSC (Yu & Da, 2023), TIOE-Det (Ming et al., 2023), and ReDet (Han et al., 2021). These works extend traditional object detection using rotated bounding boxes and are applied in aerial imagery (Xia et al., 2018), scene text (Ma et al., 2018), and industrial inspection (Wu et al., 2022). These methods typically involve deep networks requiring prolonged training time and random shuffling of many data samples. For near-ground navigation, landmark objects have fixed orientations relative to their surroundings, making the horizontal direction more critical than in-plane rotation.

Insects can remember landmark orientations after a single view without storing image data (Jeffery et al., 2016). This ability is attributed to sparse coding in neural circuits like the MB (Pearce & Bouton, 2001). The MB of *Drosophila* has been closely observed, 3D reconstructed, and its connectome analyzed (Li et al., 2020b), revealing how it processes sensory inputs via projection neurons

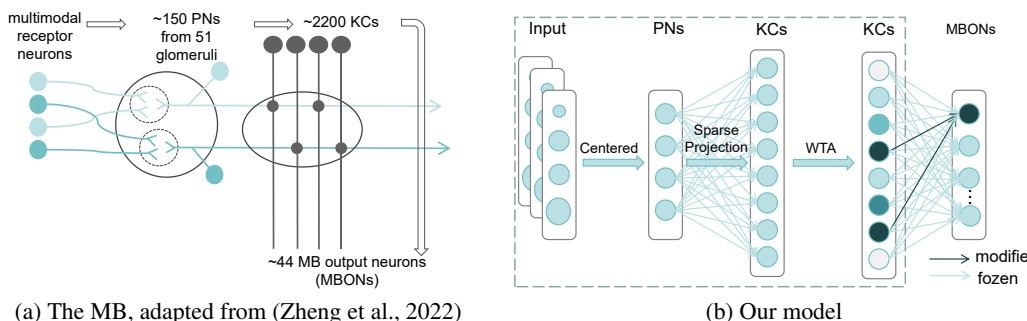

(a) The MB, adapted from (Zheng et al., 2022)     (b) Our model

Figure 1: Schematics diagram of the MB and the simplified MB model in FlyOrien. (a) The MB of a larval fruit fly *Drosophila melanogaster*, illustrating connections from sensors to the MB output neurons. (b)The simplified MB model in FlyOrien. The dashed line frames the parts for random feature extraction. Only weights between this part and "MBONs" are adjust during learning.

(PNs) to Kenyon cells (KCs) in a sparse manner (Hallem & Carlson, 2006; Stevens, 2016; Olsen et al., 2010). Only a small fraction of KCs fire simultaneously due to inhibitory feedback from the anterior paired lateral neuron (APL) (Caron et al., 2013), enabling efficient encoding and reducing interference during learning (Aso et al., 2014). The schematic of the MB is shown in Figure 1a.

Previous models have explored the MB's role in olfactory associative learning (Wessnitzer et al., 2007; Smith et al., 2008; Bennett et al., 2021). Computational neuroscience suggests the MB is crucial for insect navigation, such as visual homing (Webb & Wystrach, 2016). Visual inputs to the MB come from visual projection neurons (VPNs) and local visual interneurons (Ganguly et al., 2024; Li et al., 2020a). Models by Ardin et al. (2016) and Zhu et al. (2020) demonstrate how insects use the MB for navigation by associating visual scenes with familiar directions.

The MB's architecture has inspired computational models like FlyLSH (Dasgupta et al., 2017) for Locality Sensitive Hashing, which uses random projections and sparse coding similar to PNs and KCs (Caron et al., 2013; Baltruschat et al., 2021; Hayashi et al., 2022). The schematic plot of FlyLSH is presented in the dashed line zone of Figure 1b.

Another essential navigation circuit is the central complex (CX) (Honkanen et al., 2019), which forms a ring attractor and encodes heading and homing directions (Wu et al., 2016; Zhang, 1996). The CX integrates multiple directional cues to improve navigation accuracy (Heinze, 2017). Neuron activities predicted by computational models with ring attractors match biological observations. The connectome shows extensive connections between MB output neurons (MBONs) and the CX (Li et al., 2020a), suggesting coordination between familiarity encoding in the MB and continuous decision-making in the CX.

Inspired by the MB and CX, we propose FlyOrien, a model for incremental learning of the relative direction between an observer and an object from side-view images. We also propose biologically plausible learning rules that enable one-shot and incremental learning, reducing training time and computational resource requirements. Unlike CNNs, FlyOrien (1) does not have convolutional layers, (2) employs a wide coding layer with random, untrained weights for sparse coding, and (3) uses a learning rule minimizing interference during learning.

We demonstrate FlyOrien's effectiveness on a modified object orientation dataset and a real-world robotic orientation task. Experiments show that FlyOrien is more efficient than traditional artificial neural networks, as it only needs a single epoch training to achieve Top-5 accuracy comparable to CNNs that typically converge after 100 epochs.

The paper is structured as follows: Section 2 introduces the details of the model, and Section 3 presents the experiments, including those with a modified dataset (Section 3.1) and data from a robot in a real-world environment (Section 3.2).

## 2 MODEL

Our model, or FlyOrien, consists of two parts: a simplified MB model with firing-rate neurons and a modified associative learning rule, and a simplified CX modeled with a modified CANN. The former can learn the orientation of multiple objects, more specifically, associating a view of an object with an orientation angle. The latter merges multiple outputs of the former and provides a finer output. We also proposed a biologically plausible learning rule so that the MB model can learn images by only looking at them once.

For convenience of application, we simplified the MB and CX for a minimal model functioning in learning object orientation. It ignores neuron's morphology, uses firing-rate neuron models instead of spiking neuron models, ignores dynamics inside neurons, and treats synapses between neurons as a linear mapping. However, there are still neural dynamics by neuron interactions in the simplified CX and synaptic plasticities by a biologically plausible learning rule from KCs to MBONs in the simplified MB.

### 2.1 SIMPLIFIED MUSHROOM BODY MODEL

The simplified MB has three layers including projection neurons (Figure 1b). The first layer consists of "PNs" conveying preprocessed images. The second layer consists of "KCs" encoding images. The third layer consists of "MBONs" outputting the likelihood of angles.

#### 2.1.1 DATA PREPROCESSING

Insect sensory inputs are preprocessed before sending to KCs by PNs. The preprocessing can involve dimension reduction, noise reduction, normalization, and gain control (Gopfert & Robert, 2002). The actual preprocessing of visual signals in insects can be complex. The neural circuits in the optic lobe play an important role in processing vision in moving (Mauss et al., 2017), then visual information is projected to the MB by posterior lateral protocerebrum PNs ($_{PLP}$PNs) (Li et al., 2020c). Despite this, previous models suggest that the architecture of the mushroom body (MB) can process and learn from images without the need for complex feature extraction but directly on pixel-level information(Ardin et al., 2016; Dasgupta et al., 2017).

As a simple approximation to the optic lobe, which adjusts contrast through lateral inhibition, the first step of our model normalizes inputs. After normalization, the mean pixel intensity of each image is set to $0$. The image is then flattened to allow for the model's use across different modalities. Given a dataset $(X, y)$, where $X \in \mathbb{R}^{n \times d}$, each row represents a sample $\mathbf{x} \in \mathbb{R}^d$, $n$ is the number of sample points, and $d$ is the dimension of a sample point. A sample is shifted by the mean value $\bar{x}$ of $x$ before being passed to the PNs:

$$\widehat{\mathbf{x}} = \mathbf{x} - \bar{x}, \tag{1}$$

where $\bar{x} = \sum_{i=0}^{d} x_i/d$, $i$ is an index for the sample dimensions.

#### 2.1.2 NETWORK ARCHITECTURE

FlyOrien uses a simplified PN-KC connection and WTA for encoding samples. The synaptic weights from PN to KC are noted as a matrix $W_{PK} \in \mathbb{R}^{q \times d}$, where $q$ is the number of "KCs". The elements of $W_{PK}$ are random and binary, following a Bernoulli distribution, that is, $w_{PKji} \sim \text{Bernoulli}(p)$, where $j$ is the index of "KC" and $p = b/d$ is the probability of connection and $b$ represents the expectation of how many "PNs" are connected to a "KC". In our experiments, $b$ is set to $0.1d$ so that $p = 0.1$. With $W_{PK}$, the input to "KCs" follows:

$$\mathbf{z} = W_{PK}\widehat{\mathbf{x}}, \tag{2}$$

In the MB, the APL neuron induces lateral inhibition on KCs, allowing only the most strongly activated KCs to become active. FlyOrien approximates this WTA mechanism by keeping top $h$ activating "KCs" retain their output values, while others are set to zero:

$$\widehat{z}_j = \begin{cases} z_j & \text{if } z_j \text{ is one of the } h \text{ largest entries in } \mathbf{z}_i \\ 0 & \text{otherwise} \end{cases} \tag{3}$$

where $h$ directly controls the sparseness of the coding and $j$ is a local index here for which "KC". In our experiments, $h = 0.05q$. After WTA, the output of "KCs" is $\widehat{\mathbf{z}} = (\widehat{z}_1, \widehat{z}_2, \ldots, \widehat{z}_j, \ldots, \widehat{z}_q) \in \mathbb{R}^q$.

Since a "KC" that is always active provides little useful information, we implemented a threshold to disable such "KCs". The threshold we used is 0.25, meaning that if a "KC" remains active in more than one-quarter of the images, its output is always 0.

The synaptic weights from "KCs" to "MBONs" are presented as a matrix $W_{\mathrm{KO}} \in \mathbb{R}^{m \times q}$, where $m$ is the number of "MBONs". The activities of "MBONs" are:

$$\widehat{\mathbf{y}} = W_{\mathrm{KO}}\widehat{\mathbf{z}}, \tag{4}$$

The activity of each "MBON" is the likelihood of corresponding orientation given data sample $\mathbf{x}$.

### 2.1.3 LEARNING RULE

The MB is an associative learning center in insects. Associative learning is a type of classic conditioning that associates two stimuli or events. In the context of our model, the two stimuli are an image sample and the object orientation on the image. From an aspect of view in machine learning, we can interpret associative learning as supervised learning. Insects can continuously associate sensory stimuli with valences or behaviors, and the connections between KCs and MBONs play an important role in this process. In our model, learning occurs solely through adjusting the weights $W_{\mathrm{KO}}$ between these two layers.

We applied two variations of Hebbian rule (Hebb, 1949) for updating $W_{\mathrm{KO}}$, which are referred to as Method 1 and 2, respectively. Method 1 treats learning as a progress to converge and adjusts a weight multiple times, while Method 2 treats the learning as an instant progress and a weight can only be adjusted once. In both methods, all weights between "KCs" and the "MBONs" are initialized to 0. During training, when an image $\mathbf{x}$ and a label $y$ is provided, $\mathbf{x}$ is sparsely coded by the "KCs" as $\widehat{\mathbf{z}}$, and $y$ is presented by corresponding "MBONs" in a one-hot manner.

With a method 1, for each $\mathbf{x}$ and $y$, every activating "KC" and the "MBON" connects according to the activity of the "KC":

$$w_{\mathrm{KO}kj} = \begin{cases} \alpha_{kj}(\widehat{z}_j - w_{\mathrm{KO}kj}) + w_{\mathrm{KO}kj} & \text{if "MBON" } k \text{ is the label and "KC" } j \text{ actives,} \\ w_{\mathrm{KO}kj} & \text{otherwise.} \end{cases} \tag{5}$$

where $w_{\mathrm{KO}kj}$ is the weight from $j$th "KC" to $k$th "MBON", $\alpha_{kj}$ is the learning rate, which typically starts from 1 and decays according to the rule $\alpha_{kj} = (1 - 10^{-4})\alpha_{kj}$ if the corresponding synapse is updated. Please note that weights from inactivating "KCs" are not updated. Learning ends when all images are looped once.

Different from Method 1, Method 2 updates $W_{\mathrm{KO}}$ in a binary manner. More specifically, for each $\mathbf{x}$ and $y$, weights between activating "KCs" and the corresponding "MBON" are set to 1.

$$w_{\mathrm{KO}kj} = \begin{cases} 1 & \text{if "MBON" } k \text{ is the label and "KC" } j \text{ actives,} \\ w_{\mathrm{KO}kj} & \text{otherwise.} \end{cases} \tag{6}$$

Hence, there is no mechanism to weaken weights in Method 2. In other words, there is no forgetting on a synaptic level.

The output of the above half model is the likelihood of multi-class labels. This part of the model was evaluated in the experiment with and without the second half.

### 2.2 CANN WITH MULTIPLE INPUTS

Unlike a typical multi-class classification dataset where there are no correlations between labels, our dataset exhibits correlations between labels, allowing outputs from "MBONs" to be interpolated for finer orientation resolution. As reviewed in the introduction, in insects, multiple MBONs nerves to the fan-shaped body in the CX. As CANN has been proved to be a simplified model of CX, we built the second half of FlyOrien by modifying CANN to receive multiple outputs from "MBONs"(Fig 2).

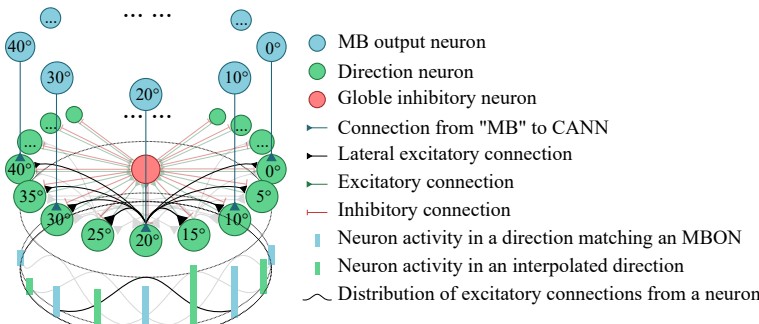

Figure 2: In our model, the continuous Attractor Neural Network (CANN) is functional as a lower-pass spatial filter and interpolator of the "MB" outputs. The "MB" outputs are fed to corresponding neurons in the CANN, which has neurons representing finer directions. The neurons, their lateral exhortatory connections, and global inhibitory connections form a ring attractor together.

The CANN for CX describes a ring attractor by multiple interconnected neurons. Every neuron is allocated with an orientation, stimulates neurons nearby and inhibits all neurons. Their input dynamics is denoted as $U(o, t)$ and described based on the orientation $o$ instead explicitly by neurons:

$$\tau \frac{\partial U(o, t)}{\partial t} = -U(o, t) + \rho \int_{x'} J(o, o') r(o', t) dx' + I^{ext}(o, t) \tag{7}$$

Where $\tau$ is the time constant for the population dynamics, which is on the order of $1$ms (Gutkin et al., 2003), $\rho = h/(2\pi)$ is the neural density and $h$ is the number that orientation is discretized, $I^{ext}(o, t)$ is the input to the neuron at $o$ at time $t$. $J(o, o') = \frac{J_0}{\sqrt{2\pi}a} \exp(-|o - o'|^2/2a^2)$ presents the excitatory connections from the neuron at $o'$ to the neuron at $o$, where $a = 0.1$ is the half-width of the range of excitatory connections. $r(o, t)$ is the firing rate of neurons:

$$r(o, t) = \frac{U(o, t)^2}{1 + k\rho \int U(o', t)^2 do'} \tag{8}$$

where $k = 0.1$ is the degree of the inhibition. The contribution of inhibitory connection is achieved indirectly through the divisive normalization in equation 8.

The output of the simplified MB model is fed to the CANN by the term $I^{ext}(o, t)$, where $o$ corresponds to the labels of "MBONs". As shown in Fig 2, there are more neurons in CANN than MBONs for finer directions, and each MBON outputs to a corresponding neuron for the same direction. Thus with the dynamics of CANN, CANN can integrate information from multiple outputs from the "MBONs", and predict finer orientations.

Thus, we can add more neurons in CANN to interpolate for a finer resolution output. The model is implemented with Python and attached in supplementary material.

## 3 EXPERIMENTS

We tested the model on a dataset for object orientation learning and a dataset from a robot for real-world evaluation. There are two types of tasks: retrieval and prediction. Please note the retrieval task tests the ability of the models to associate images with their corresponding orientations, thus the same images are presented in the test. Computation is conducted on a desktop workstation with the 12th Gen Intel ® Core ™i7-12700 Processor, 32GB RAM, and the NVIDIA® GeForce RTX ™3090. We compared our model with typical convolutional neural networks (CNNs) in object orientation retrieving and prediction, and our model trained on CPU can even achieve better performance 7 to 45 times faster in training time than CNNs trained in GPU.

### 3.1 OBJECT ORIENTATIONS LEARNING IN COIL DATASET

Our model was evaluated on a dataset modified from COIL-100 dataset (Nene et al., 1996) along with baseline models. The original dataset contains 100 objects captured at 72 different orientations

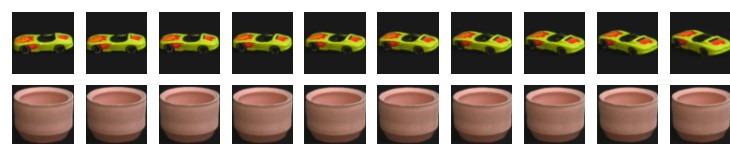

Figure 3: Example samples from COIL-100-O (Top) and COIL-100-AS (Bottom).

and in total 7200 images which are labeled with the object. The size of the original image is $128 \times 128$, for each of the images, there are $128 \times 128 \times 3 = 49152$ channels of values as the image is RGB colored. We modified the dataset by associating the images with object orientation instead of the object. Thus, different objects can associated with the same label, while the same objects are labeled differently, and there is strong interference while a model is trained on this modified dataset.

Because there is no correlation between samples with the same label in this dataset, cross-validation is unsuitable for this task. This is a key distinction from typical datasets. In most classification tasks, samples with the same label share similar features, allowing for knowledge generalization across those samples. However, this is not the case in our dataset. Since samples with the same label are not correlated, cross-validation, which typically evaluates generalization within samples of the same label, becomes less meaningful. As we will show later, both baseline models and our proposed model have achieved near-zero accuracy with cross-validation (Figure A3, Table A6).

We divided this dataset into two groups according to whether the object is axisymmetric and without a textured pattern, resulting in COIL-100-Ordinary(COIL-100-O) group and COIL-100-Axisymmetric(COIL-100-AS) group. For COIL-100-O, the objects are not axisymmetric or have clear textured patterns. For COIL-100-AS, the objects are axisymmetric without views of a clear textured pattern. In COIL-100-AS, different views of the same object are so similar that human eyes cannot even distinguish them. We present views of two objects (Figure 3), the first row is from COIL-100-O, and the second row is from COIL-100-AS.

### 3.1.1 RETRIEVAL TASK BY THE SIMPLIFIED MB

The first experiment on COIL is the retrieval of object orientation. This experiment does not discriminate between the training set and the testing set. Instead, the model should retrieve the angle of objects in the previously viewed image. It is conceptually simple, but because the same object shares the same features but has different labels for orientation, there is interference when a typical ANN learns the orientations. The Top-5 criterion is applied to retrieval accuracy. That is, if the correct label is in the Top-5 predicted labels by a model, this model predicts correctly.

In training, the learning rules proposed in Section 2.1.3 were applied to our model. With the learning rule, our model only loops through the dataset once. Differently, baseline models were trained for 200 epochs. They were optimized with the Adam optimizer implemented in PyTorch with default parameters. The loss function for gradient descent was cross entropy provided by PyTorch with default parameters.

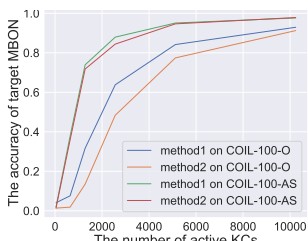

Figure 4: Accuracy with different number of KCs.

**More KCs, more accurate.** We evaluated the influence of the number of active KCs on retrieval accuracy. As the number of KCs increases, the accuracy of our methods improves across both datasets, approaching convergence when the number of KCs is close to 10,000 (Figure 4).

**Retrieval accuracy of the simplified MB** As our model's performance converges around 10,000 KCs, we used models with 10,240 KCs for comparison with the baselines. This choice is in favor of common multiples of powers of 2 and 10 and also aligns with biological plausibility (Abdelrahman et al., 2021). Figure A1 and A2 show the Top-5 active MBONs for every object in an example orientation in COIL. The first column is an example orientation, the second column is the corresponding Top-5 MBONs with weights learned by method 1, and the third column shows the results from method 2. FlyOrien achieves more than 90% accuracy across both datasets in retrieving the orientation of a viewed object after a single learning instance.

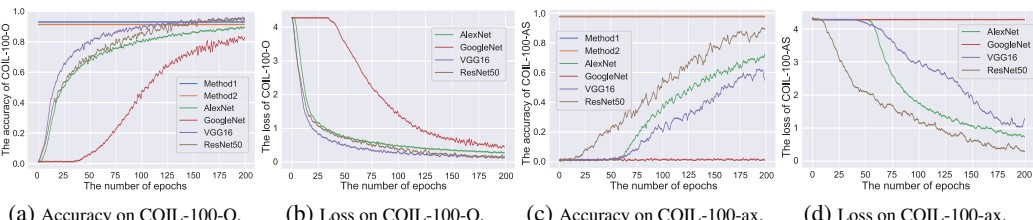

(a) Accuracy on COIL-100-O.  (b) Loss on COIL-100-O.  (c) Accuracy on COIL-100-ax.  (d) Loss on COIL-100-ax.

Figure 5: Accuracy and loss in the retrieval task on COIL-100-O and COIL-100-ax.

**Baselines take much longer training time for the same performance.** We compared the accuracy, training time, and incremental learning ability of our two methods with CNNs like AlexNet (Krizhevsky et al., 2012), GoogleNet (Szegedy et al., 2015), VGG16(Simonyan & Zisserman, 2014), ResNet50 (He et al., 2016), as illustrated in Table 1. The accuracy and loss change of increasing epochs for the baselines is shown in Figure 5. In Figure 5a and 5c, our methods are displayed as horizontal lines because they only need to be learned once. Other models take 1.6 to 80.6 times longer for a similar performance. Please note that we did not accelerate our model on GPU.

Table 1: Retrieval accuracy (%) and training time (s) of the simplified MB and baselines.

| Method | Platform | COIL-100-O | | COIL-100-AS | |
|---|---|---|---|---|---|
| | | Acc | Time | Acc | Time |
| Method 1 | CPU | 92.93 | 112 | 97.65 | 47 |
| Method 2 | CPU | 91.26 | 61 | 97.86 | 47 |
| AlexNet | GPU | 97.77 | 873 | 86.22 | 131 |
| GoogleNet | GPU | 92.77 | 1845 | 35.01 | 273 |
| VGG16 | GPU | 97.91 | 10390 | 71.05 | 1537 |
| ResNet50 | GPU | 97.92 | 4317 | 95.30 | 639 |
| MobileNet | GPU | 99.89 | 947 | 79.81 | 166 |
| Shufflenet | GPU | 99.51 | 1651 | 83.55 | 289 |

**Incremental learning ability** We trained FlyOrien incrementally and calculated accuracy on previously trained objects to assess the model's incremental learning ability. Specifically, after training on all images of an object, we evaluate the model's accuracy on every object that has been learned. The results, shown in Appendix Figures A4 to A5, indicate that our model can acquire new knowledge without forgetting previously learned knowledge, even for axisymmetric objects that are challenging for humans. Appendix Figure A6 shows the results in the dimension of time along with results by baseline models in an incremental learning setup. It demonstrates that while all baseline models experience catastrophic forgetting over 10 iterations of optimization, our model is nearly unaffected by the trained order of samples.

### 3.1.2 PREDICTION ACCURACY OF THE SIMPLIFIED MB AND CX

The first half of FlyOrien outputs label likelihoods in a multi-class classification setup. However, for real-world applications, we aim for more precise predictions. This experiment evaluates the capability of the full FlyOrien model, combined with CANN, to predict orientations with a finer resolution than that used in training. For ease of evaluation, we divided the data based on object orientations, with 72 evenly distributed orientations, alternating between the training and testing sets.

For a fair comparison, we also integrated the baseline models with CANN, resulting in two setups: models with and without CANN. In the first setup, without CANN, orientations in the testing set cannot be predicted directly, so the adjacent angle is used as the correct prediction criterion. In the second setup, although the baseline models only predict orientations in the testing set, with CANN, the orientations in the training set can be predicted, so the Top-5 criterion for multi-class classification is applied. It is important to note that the evaluation criteria differ between these two setups, and comparisons are valid only within the same setup.

With the first setup and Method 2, the simplified MB in our model outperforms baselines (Table 2, second and third rows). The accuracy of the simplified MB is 95.34% on the testing set while the best baseline is AlexNet with 91.28% accuracy. With the second setup and Method 2, the simplified MB with CANN, or the full FlyOrien model, has the highest training accuracy 98.95%, while not best for testing accuracy, 66.57%. A possible reason is that the simplified MB tends to output a bimodal distribution, and there is a second set of large likelihood peaks on the opposite side of the orientation, which moves the peak of CANN away from the correct orientation.

Table 2: Accuracy (%) of FlyOrien and four baselines.

| Model | Original Model | | Original Model + CANN | |
|---|---|---|---|---|
| | Training | Testing | Training | Testing |
| Simplified MB | 98.86 | 95.34 | 98.95 | 66.57 |
| AlexNet | 92.27 | 91.28 | 83.88 | 20.34 |
| GoogleNet | 11.59 | 2.78 | 3.35 | 5.56 |
| VGG16 | 94.41 | 90.84 | 90.33 | 81.45 |
| ResNet50 | 94.06 | 90.94 | 85.82 | 81.96 |
| MobileNet | 95.09 | 94.44 | 78.21 | 77.56 |
| Shufflenet | 71.37 | 79.06 | 55.77 | 54.49 |

We also evaluated the training accuracy and testing accuracy of our model and baseline models on the COIL-100-O dataset with altered contrast in images. For more details, see Section A.2.5.

## 3.2 DETECTION ACCURACY ON REAL OBJECTS COLLECTED BY A QUADRUPED ROBOT

To simulate an animal finding directions, two experiments were conducted on a quadruped robot. In experiment 1, the robot finds a familiar object or landmark in the environment and makes an angle judgment around the landmark 360°. In the second experiment, in the empty scene with no suitable objects or landmarks to be surrounded, the angle is according to its own orientation. Each sample collected a total of 360 images, with each image size of 128 x 128 pixels. Compared with the test of re-designed datasets, the angle interval of the testing set of this experiment has changed from 10° to 1°, which is more dense and more consistent with the randomness of the angle and position of the robot in the real scene.

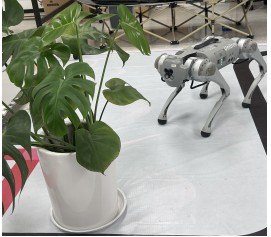

Figure 6: A quadruped robot looks at a landmark.

**Object's orientation by vision from robot**

In this experiment, a robot motion control and image sampling algorithm was designed to realize the robot sampling from different angles during the 360° rotation around the object (Figure6). The robot rotates around an object, taking one photo per degree with its head camera, for a total of 360 photos. The binocular fisheye cameras on the robot's head have a 180° field of view. Through the official camera calibration algorithm built into the robot, the corrected photos are transmitted in real time during the sampling process. The image from both the left and right eyes is 800×928. We will extract the image from the left eye for use in the subsequent experiment, compressing it to 128×128.

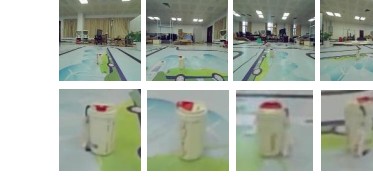 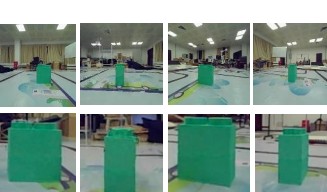 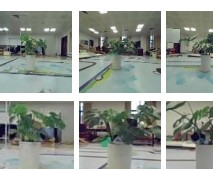
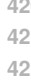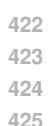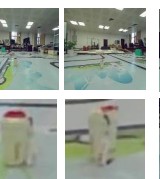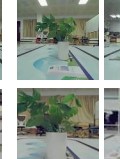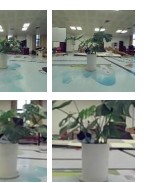

Figure 7: Sampled images of a cup, a foam box, and a plant at 0°, 90°, 180°, and 270° (Top: original view from the robot, Bottom: cropped view on objects for orientation.)

The photos of every ten degrees are selected as the training data set, the rest are taken as the testing set, and the nearest ten degrees are taken as the label for the accuracy test. Cups, foam boxes, and

Figure 8: Sampling images of lab1, lab2, and corridor at 0°, 90°, 180°, and 270°

plants were sampled and tested for accuracy. The uncropped data sets were also tested (Figure7). It can be observed that the robot's Top-5 object orientation accuracy is over 96%, and the Top-2 accuracy is over 80% (Table 3). The performance remains relatively stable on the dense testing set with 1° intervals.

Table 3: Accuracy of network trained with a single data set.

| Object | cup | foam box | plant | cup (in scene) | foam box (in scene) | plant (in scene) |
|--------|-----|----------|-------|----------------|---------------------|------------------|
| acc(Top-2) | 80.56 | 82.93 | 83.33 | 88.89 | 87.83 | 91.11 |
| acc(Top-5) | 97.78 | 97.56 | 96.67 | 98.89 | 97.72 | 98.89 |

**Robot's orientation by vision to the environment**

In this experiment, the robot can rotate in circles only by setting rotational speed. The robot was made to rotate itself once for sampling in three different scenes: lab 1, lab 2, and corridor(Fig.8). The sampling method is the same as in the previous experiment. The robot achieved an orientation accuracy of over 96% when choosing Top-5 activated MBONs, and over 80% when choosing Top-2 activated MBONs (Table 4). The performance remains relatively stable on the dense testing set with 1° intervals.

| Scene | lab1 | lab2 | corridor |
|-------|------|------|----------|
| acc(Top-2) | 91.41 | 87.52 | 83.17 |
| acc(Top-5) | 98.53 | 99.31 | 93.32 |

| Original Method | dataset 1 | dataset 2 | dataset 3 |
|-----------------|-----------|-----------|-----------|
| acc(Top-2) | 77.31 | 79.72 | 79.14 |
| acc(Top-5) | 96.00 | 88.98 | 95.13 |

Table 4: Accuracy on a single dataset  Table 5: Accuracy on a complex situation

**Training network testing on complex data sets**

The six image datasets from the Object's orientation experiments were combined into Dataset 1. The three image datasets from the Robot's orientation experiments were combined into Dataset 2. Finally, Dataset 1 and Dataset 2 were merged into Dataset 3 to test the neural network's stability in long-term learning. By comparing Table5 with Tables 3 and 4, and by comparing Dataset 3 with Dataset 1 and Dataset 2, it can be observed that the accuracy of neural network was not significantly affected by the change in the data set from single to complex. This indicates that the neural network has good stability for long-term learning.

## 4 DISCUSSION AND CONCLUSION

Inspired by the neural circuits of insects, particularly the MB and CX, we proposed FlyOrien, a bio-inspired model for incremental learning of object orientation. The model mimics the MB's sparse coding and associative learning while utilizing the CX to integrate multiple sensory inputs to refine orientation detection. FlyOrien is designed to learn object orientations efficiently after a single exposure, and because it mimics the sparse coding of MB, it has the potential to generalize to multimodal inputs, such as posture, olfactory, and directional cues, which will be investigated in further research.

FlyOrien was tested on open-source datasets and real-world robotic tasks, demonstrating strong performance in estimating object orientations and handling ego motion in complex scenes. Its ability to learn incrementally, without large datasets or extensive training, highlights its suitability for real-time applications.

Without relying on convolutional layers, FlyOrien learns object orientation efficiently without catastrophic forgetting, benefiting from its large number of pattern detectors and sparse coding. For instance, samples in COIL-100-AS with the same label are very similar, so subtle features, such as

specific patterns, are crucial for orientation detection, but CNNs are not optimized for this. CNNs generalize by learning from fewer images and using shared weights to capture relationships between local features. Convolutional kernels in the first layer detect low-level features, but this generalization can overshadow rare or unique patterns, risking them being forgotten. In contrast, MB-like architectures excel at identifying these subtle features and preventing forgetting by maintaining fixed connections after learning. In our model, many "KCs," each connected to only a few pixels, act as specialized pattern detectors. Unlike CNNs, which apply the same filters across regions, FlyOrien uses more filters simultaneously, detecting intricate details in a single pass. This key difference enables FlyOrien to perform better and learn faster in our tasks.

While FlyOrien offers significant benefits, it is sensitive to pixel-level changes, affecting performance when objects deform or lighting varies. Addressing these limitations is a key area for future research, particularly by incorporating the optic lobe which is crucial for dynamic vision processing. Extending the CX model to a two-dimensional CANN could also improve navigation in complex, unmapped environments, enhancing FlyOrien's robustness for more sophisticated spatial tasks.

FlyOrien's lightweight design, free from GPU dependence, allows it to run effectively on small devices like drones and robots, making it ideal for resource-constrained tasks like object tracking, navigation, and surveillance, where low power consumption and computational efficiency are critical.

In practical applications, FlyOrien presents minimal risks. Its use in autonomous robots can improve navigation and object recognition without needing extensive computational resources. However, ensuring transparency and human oversight in deployment is crucial. When used for navigation or surveillance in public spaces, it's important to respect privacy and operate within ethical guidelines. FlyOrien's efficiency on small robots makes it ideal for search and rescue, environmental monitoring, and industrial automation. With safeguards in place, FlyOrien can positively contribute to these fields without significant risks.

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

# A  APPENDIX

## A.1  ALGORITHM

### A.1.1  ALGORITHM FOR DATA PREPROCESSING AND NETWORK ARCHITECTURE

---

**Algorithm 1** Data Preprocessing and Network Architecture in the simplified MB

---

1: **Input:** Dataset $(X, y)$ where $X \in \mathbb{R}^{n \times d}$
2: **Output:** Activation of "MBONs" $\widehat{\mathbf{z}}$ for labels' likelihood
3: Initialize weights $W_{\mathrm{PK}} \in \mathbb{R}^{q \times d}$, $w_{\mathrm{PK}ji} \sim \mathrm{Bernoulli}(p)$
4: Initialize "KCs" activation mask: $\mathbf{v} = \mathbf{1}$
5: Initialize weights $W_{\mathrm{KO}} \in \mathbb{R}^{m \times q}$
6: // Step 1: Normalize Inputs
7: **for** each sample $\mathbf{x} \in X$ **do**
8:     Compute mean $\bar{x} = \frac{1}{d} \sum_{j=1}^{d} x_j$
9:     Shift the sample: $\widehat{\mathbf{x}} = \mathbf{x} - \bar{x}$
10: **end for**
11: **for** each sample $\mathbf{x} \in X$ **do**
12:     // Step 2: Activation and outputs of "KCs"
13:     Compute "KCs" activation: $\mathbf{z} = W_{\mathrm{PK}}\widehat{\mathbf{x}}$
14:     Keep top $h$ activating KCs, whose indexes are entries of $\mathbf{u}$
15:     **for** each $j \in \mathbf{u}$ **do**
16:         **if** $z_j$ is one of the $h$ largest entries in $\mathbf{z}$ **then**
17:             $\widehat{z}_j = z_j v_j$
18:         **else**
19:             $\widehat{z}_j = 0$
20:         **end if**
21:     **end for**
22:     // Step 3: Optionally disable over-activating "KCs"
23:     **for** each $j \in \mathbf{u}$ **do**
24:         **if** KC $j$ response to more than $1/4$ samples  **then**
25:             $v_j = 0$
26:         **end if**
27:     **end for**
28:     // Step 4: Activation and outputs of "MBONs"
29:     Compute "MBONs" activities: $\widehat{\mathbf{y}} = W_{\mathrm{KO}}\widehat{\mathbf{z}}$
30:     // Step 5: Learning Rule
31:     **Method 1: Hebbian Learning with continuous Weights**
32:     **for** each active KC $j$ **do**
33:         **if** $k$ is the label $y$ **then**
34:             Update weights: $w_{\mathrm{KO}kj} \leftarrow \alpha(\widehat{z}_j - w_{\mathrm{KO}kj}) + w_{\mathrm{KO}kj}$
35:         **end if**
36:     **end for**
37:     Decay learning rate: $\alpha \leftarrow (1 - 10^{-4})\alpha$
38:     **Method 2: Hebbian Learning with Binary Weights**
39:     **for** each active KC $j$ **do**
40:         **if** $k$ is the label $y$ **then**
41:             Set weight: $w_{\mathrm{KO}kj} \leftarrow 1$
42:         **end if**
43:     **end for**
44: **end for**

---

## A.2 Experiments

### A.2.1 Top-5 active MBONs for the whole dataset

| Object | Original angle | Top5 active MBONs(method1) | Top5 active MBONs(method2) | Object | Original angle | Top5 active MBONs(method1) | Top5 active MBONs(method2) |
|---|---|---|---|---|---|---|---|
| | 45 | 45, 30, 40, 25, 215 | 45, 30, 25, 40, 215 | | 310 | 310, 315, 320, 120, 305 | 310, 315, 320, 120, 305 |
| | 30 | 30, 35, 25, 20, 40 | 30, 35, 40, 20, 25 | | 255 | 255,265,260,270,275 | 255,265,275,260,160 |
| | 115 | 115, 110, 145, 255, 180 | 115, 145, 150, 110, 255 | | 220 | 220,215,225,230,210 | 220,215,225,230,210 |
| | 75 | 75, 80, 70, 310, 85 | 75, 80, 310, 70, 85 | | 50 | 50,55,45,65,60 | 50,55,65,45,40 |
| | 225 | 225, 45, 40, 220, 35 | 225, 40, 45, 35, 220 | | 165 | 165,185,195,160,170 | 165,185,195,170,160 |
| | 25 | 25, 20, 10,15, 30 | 425, 20, 10, 15, 35 | | 230 | 230, 255, 250, 225, 0 | 125, 215, 225, 230, 250 |
| | 80 | 80, 75, 85, 90, 70 | 80, 75, 85, 90, 95 | | 190 | 190, 195, 200, 180, 205 | 190, 195, 200, 180, 170 |
| | 160 | 160, 165, 155, 185, 150 | 160, 150, 155, 165, 185 | | 170 | 170, 165, 160, 175, 155 | 170, 165, 160, 145, 155 |
| | 165 | 165, 190, 170, 160, 195 | 165, 190, 170, 160, 200 | | 25 | 20, 15, 25, 30, 10 | 20, 15, 25, 30, 10 |
| | 130 | 130, 135, 140, 125, 120 | 130, 135, 125, 140, 120 | | 215 | 215, 210, 50, 220, 90 | 210, 215, 290, 230, 265 |
| | 220 | 220, 215, 225, 230, 210 | 220, 215, 225, 230, 210 | | 25 | 25, 10, 30, 20, 5 | 25, 10, 30, 20, 5 |
| | 205 | 205, 245, 215, 70, 220 | 205, 215, 245, 200, 220 | | 190 | 190, 185, 195, 210, 175 | 185, 190, 205, 210, 195 |
| | 60 | 60, 65, 55, 40, 15 | 60, 65, 55, 40, 15 | | 155 | 155, 175, 160, 145, 170 | 155, 175, 145, 170, 160 |
| | 45 | 45, 40, 50, 35, 30 | 45, 40, 50, 35, 55 | | 25 | 25, 10, 30, 5, 20 | 25, 10, 30, 20, 5 |
| | 170 | 170,175, 165, 180, 185 | 170, 175, 165, 180, 185 | | 145 | 145, 150, 140, 135, 130 | 145, 150, 140, 320, 130 |
| | 15 | 15, 10, 5, 20, 0 | 15, 10, 5, 20, 0 | | 55 | 55, 60, 70, 85, 80 | 55, 60, 70, 85, 40 |
| | 210 | 210, 205, 200, 215, 195 | 210, 205, 200, 195, 215 | | 125 | 125, 120, 115, 130, 135 | 125, 120, 115, 130, 135 |
| | 175 | 175, 170, 180, 165, 160 | 175, 180, 170, 165, 185 | | 185 | 185, 180, 190, 165, 175 | 185, 180, 190, 165, 175 |
| | 260 | 260, 265, 250, 255, 280 | 260, 250, 265, 285, 290 | | 170 | 170, 175, 165, 180, 0 | 170, 175, 165, 180, 185 |
| | 0 | 0, 5, 10, 50, 15 | 0, 5, 10, 15, 50 | | 35 | 35, 20, 30, 25, 40 | 35, 20, 25, 30, 15 |
| | 210 | 210, 200, 215, 190, 195 | 210, 200, 190, 215, 195 | | 80 | 80, 85, 100, 75, 90 | 80, 85, 100, 75, 90 |
| | 310 | 310, 305, 315, 300, 295 | 310, 305, 315, 295, 300 | | 45 | 45, 35, 40, 50, 10 | 45, 35, 40, 10, 50 |
| | 285 | 285, 280, 290, 265, 275 | 285, 280, 290, 275, 240 | | 15 | 15, 20, 10, 5, 25 | 15, 20, 10, 5, 25 |
| | 145 | 145, 130, 135, 175, 140 | 145, 135, 130, 90, 140 | | 80 | 80, 75, 85, 70, 45 | 80, 75, 85, 70, 45 |
| | 355 | 355, 0, 350, 15, 0 | 355, 350, 0, 10, 15 | | 150 | 150, 155, 5, 160, 25 | 150, 155, 185, 25, 160 |
| | 10 | 10, 25, 20, 15, 5 | 10, 20, 15, 25, 5 | | 130 | 130, 140, 270, 105, 110 | 130, 140, 135, 230, 125 |
| | 35 | 35, 20, 315, 30, 15 | 35, 15, 25, 30, 20 | | 325 | 325, 315, 275, 330, 335 | 325, 315, 275, 350, 335 |
| | 340 | 340, 345, 290, 95, 90 | 345, 340, 290, 350, 285 | | 295 | 295, 205, 225, 325, 245 | 295, 205,325, 330, 225 |
| | 305 | 305, 275, 295, 290, 300 | 305, 275, 310, 300, 295 | | 260 | 250, 255, 75, 235, 265 | 260, 255, 75, 265, 235 |
| | 70 | 70, 65, 75, 90, 95 | 70, 75, 65, 90, 80 | | 15 | 15, 20, 10, 5, 25 | 15, 20, 10, 5, 25 |

Figure A1: Top-5 active MBONS and the original orientations for all objects(Part1).

| | | | | | | | |
|---|---|---|---|---|---|---|---|
| | 205° | 205, 210, 215, 200,25 | 205, 210, 215, 200, 195 | | 230 | 230, 235, 220, 205, 225 | 230, 235, 220, 205, 225 |
| | 150 | 150, 170, 155, 165, 100 | 150, 155, 175, 135, 165 | | 205 | 205, 200, 210, 190, 215 | 205, 200, 210, 190, 215 |
| | 210 | 210, 215, 220, 270, 285 | 210, 215, 220, 230,225 | | 65 | 65, 70, 60, 80, 55 | 65, 60, 55, 70, 80 |
| | 170 | 170, 165, 175, 190, 180 | 170, 175, 165, 190, 180 | | 10 | 10, 15, 20, 25, 5 | 10, 15, 20, 25, 5 |
| | 45 | 45, 40, 50, 35, 30 | 45, 40, 50, 35, 30 | | 180 | 180, 185, 190, 170, 140 | 180, 185, 190, 170, 140 |
| | 25 | 25, 20, 30, 55, 10 | 25, 20, 30, 10,5 | | 175 | 175, 165, 120, 185,0 | 175, 120, 165, 180, 220 |
| | 135 | 135, 140, 145, 130, 160 | 135, 140, 145, 130, 160 | | 40 | 40, 45, 35, 50, 25 | 40, 45, 50, 35, 25 |
| | 160 | 160, 165, 175, 170, 155 | 160, 165, 175, 155, 170 | | 5 | 5, 0, 10, 15, 355 | 5, 0, 10, 15, 20 |
| | 130 | 130, 125, 310,320, 135 | 130, 125, 310, 320, 315 | | 115 | 115, 110, 105, 120, 125 | 115, 110, 120, 105, 125 |
| | 25 | 25, 20, 15, 35, 30 | 25, 20, 15, 35, 30 | | 170 | 170, 165, 160, 175, 185 | 170, 160, 165, 175, 185 |
| | 45 | 45, 70, 60, 65, 40 | 45, 40, 55, 60, 65 | | 315 | 315, 285, 280, 310, 290 | 315, 285, 320, 310, 290 |
| | 115 | 115, 110, 120, 110, 130 | 115, 120, 110, 130, 320 | | 175 | 175, 180, 170, 190, 185 | 175, 170, 180, 190, 185 |
| | 195 | 195, 200, 190, 15, 345 | 195, 200, 190, 15, 345 | | 170 | 170, 185, 145, 155, 180 | 170, 185, 145, 180, 155 |
| | 150 | 150, 145, 175, 160, 155 | 150, 145, 185, 170, 175 | | 220 | 220, 215, 180, 185, 195 | 220, 215, 180, 250, 185 |
| | 120 | 120, 135, 130, 125, 140 | 120, 130, 135, 125, 155 | | 90 | 95, 100, 80, 85, 60 | 95, 100, 85, 80, 75 |
| | 35 | 35, 30, 25, 40, 45 | 35, 30, 25, 40, 45 | | 315 | 315, 305, 285, 240, 295 | 315, 305, 325, 310, 320 |
| | 185 | 185, 190, 180, 195, 175 | 185, 190, 180, 175, 195 | | 175 | 175, 180, 170, 165, 185 | 175, 180, 170, 165, 185 |
| | 100 | 100, 95, 90, 115, 110 | 100, 95, 90, 115, 110 | | 85 | 85, 90, 100, 105, 95 | 85, 90, 100, 105, 70 |
| | 40 | 40, 45, 25, 30, 35 | 40, 45, 30, 25, 35 | | 50 | 50, 35, 25, 20, 45 | 50, 35, 45, 25, 20 |
| | 70 | 70, 75, 80, 55, 35 | 70, 75, 55, 80, 45 | | 180 | 180, 185, 190, 175, 200 | 180, 185, 190, 175, 195 |

Figure A2: Top 5 active MBONs and the original orientations for all objects(Part2).

We show all objects at an example orientation in Figure A1 and A2. The first column is the actual orientation of the object, the second column is top 5 active MBONs found by method 1, the third cloumn is method 2.

### A.2.2 CROSS-VALIDATION

| Model | Best accuracy on testing set | Best accuracy on training set |
|---|---|---|
| AlexNet | 0.00 | 92.37 |
| GoogleNet | 0.00 | 87.32 |
| VGG16 | 0.00 | 97.36 |
| ResNet50 | 0.00 | 96.53 |

Table A6: Cross-validation on our modified dataset results in 0 testing accuracy.

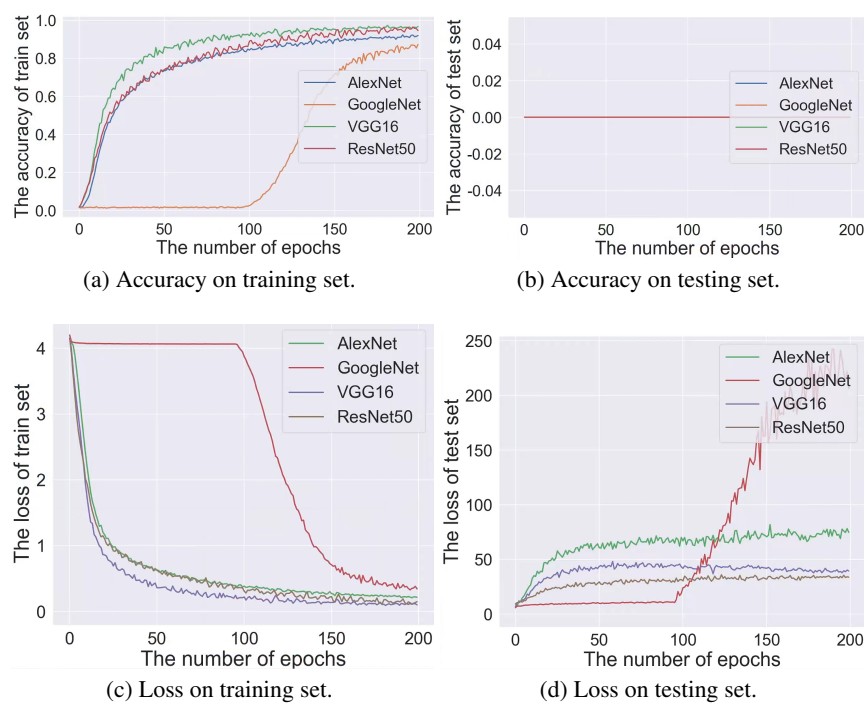

(a) Accuracy on training set.    (b) Accuracy on testing set.

(c) Loss on training set.    (d) Loss on testing set.

Figure A3: Cross-validation on our modified dataset results in 0 testing accuracy.

### A.2.3 INCREMENTAL LEARNING ABILITY

From Figure A4 to Figure A5, we show the accuracy change of learned objects when learning new objects using FlyOrien. Both method 1 and method 2 can keep good memory of old objects, thus have good incremental learning ability.

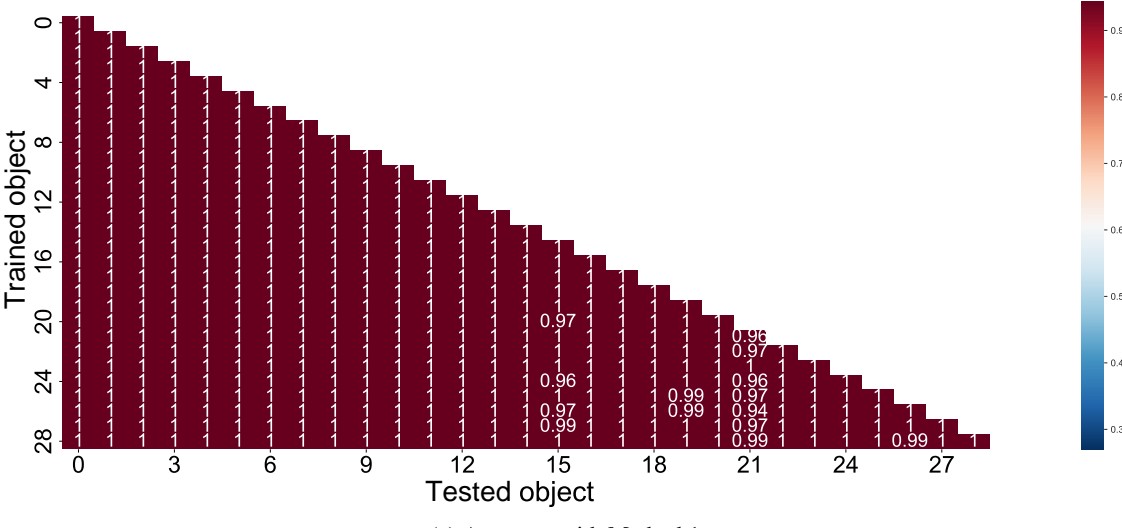

(a) Accuracy with Method 1

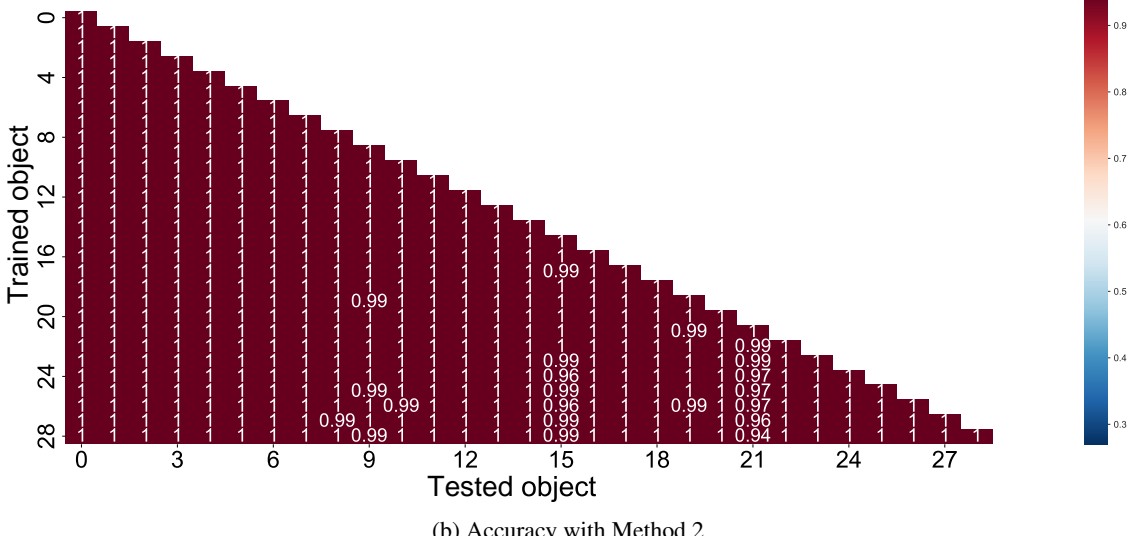

(b) Accuracy with Method 2

Figure A4: Accuracy of specific objects in COIL-100-O during incremental learning. Each row represents the index of the object being trained on, and each column represents the index of the object being retrieved. (a) Accuracy using Method 1. (b) Accuracy using Method 2. Results are shown for the first 29 objects only.

In Figure A6, we show the first four objects' accuracy change when learning new objects. We train and test the object in sequence. For deep neural networks, it lost memory of old objects when learning new objects. In contrast, FlyOrien performs well in this situation.

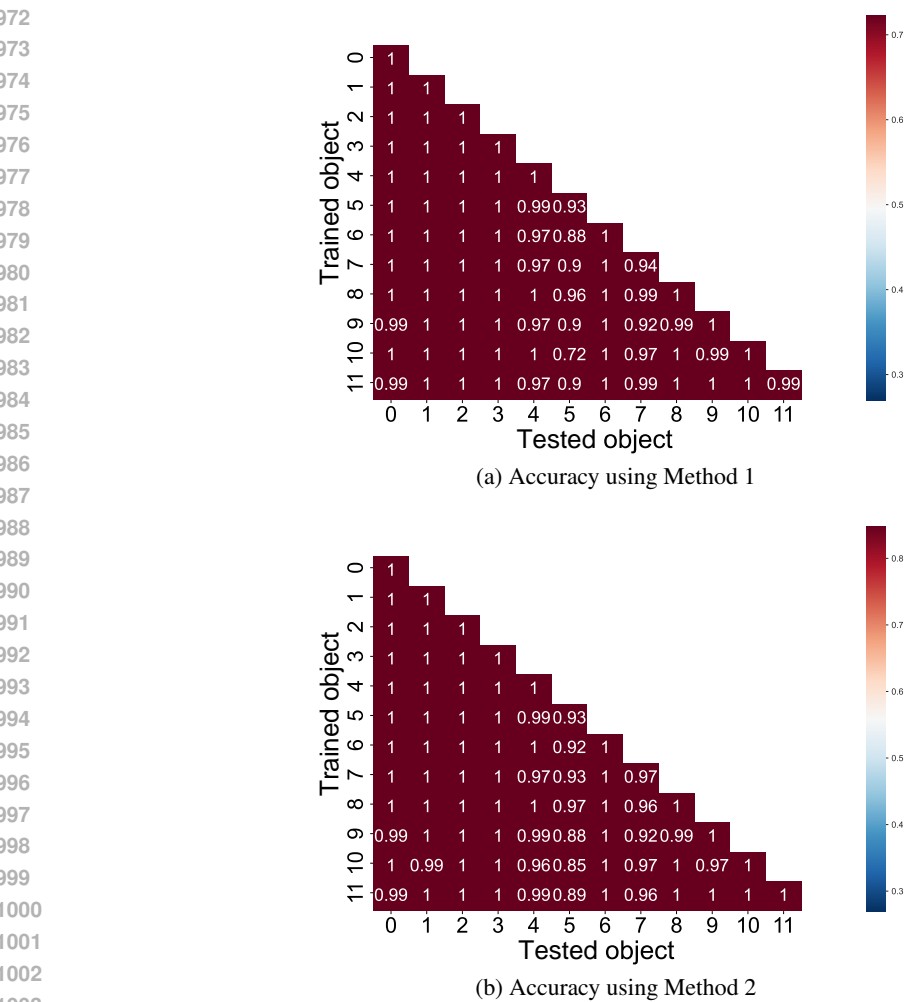

(a) Accuracy using Method 1

(b) Accuracy using Method 2

Figure A5: Accuracy of specific objects in COIL-100-AS during incremental learning. Each row represents the index of the object being trained on, and each column represents the index of the object being retrieved. (a) Accuracy using Method 1. (b) Accuracy using Method 2. Results are shown for the first 12 objects only.

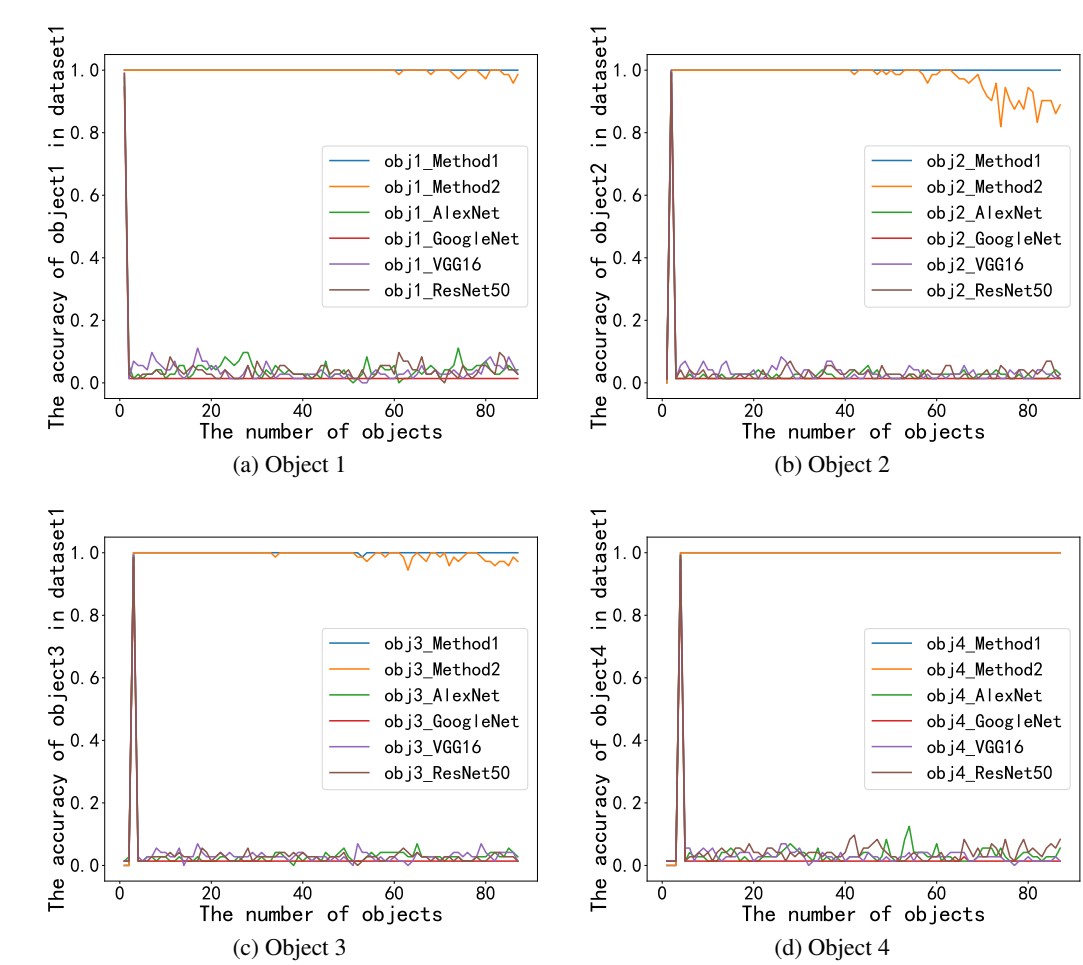

Figure A6: Accuracy of the first four objects for incremental learning.

### A.2.4 ACCURACY FOR UNFAMILIAR ORIENTATIONS

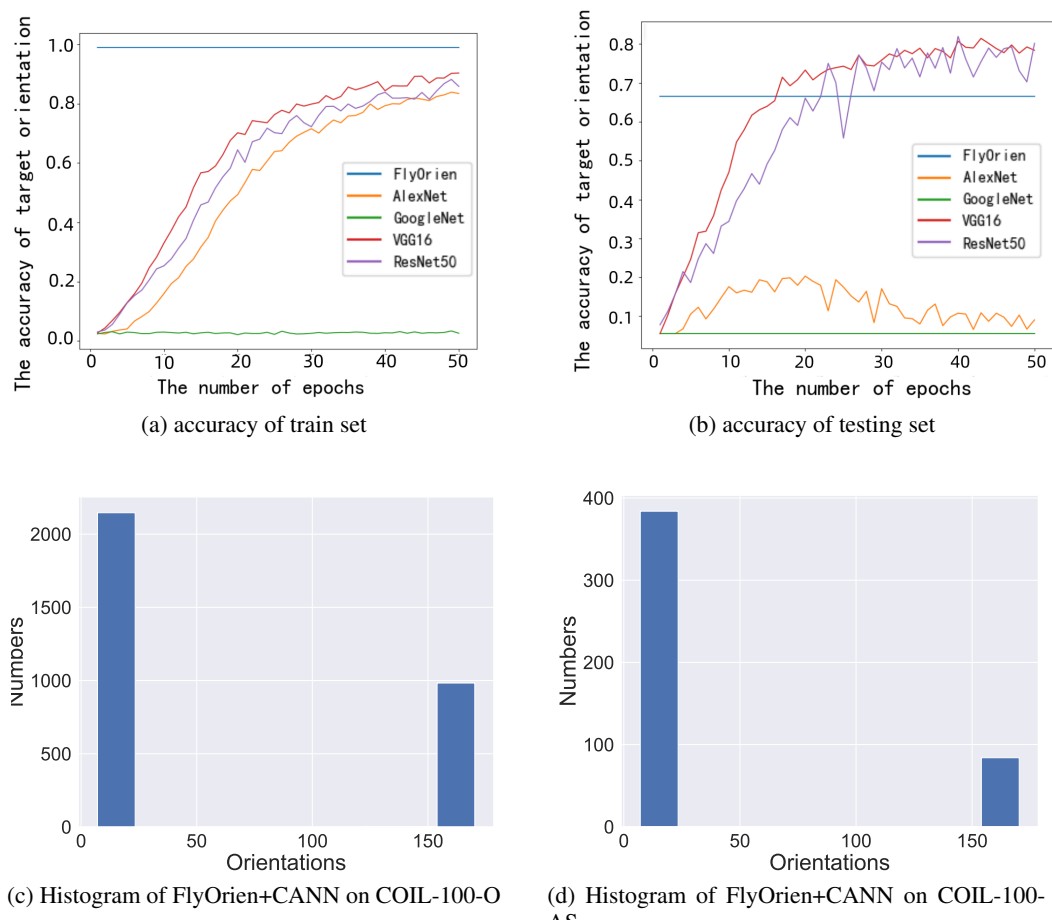

(a) accuracy of train set

(b) accuracy of testing set

(c) Histogram of FlyOrien+CANN on COIL-100-O

(d) Histogram of FlyOrien+CANN on COIL-100-AS

Figure A7: Accuracy of train set and testing set. FlyOrien's accuracy can reach a high level in one-shot learning, without longtime training like other baselines

### A.2.5 ACCURACY WITH CONTRAST CHANGES

Table A7: Accuracy(%) and training time(s) of our methods and four baselines when the image contrast changes on COIL-100-O.

|  | Method 1 | Method 2 | AlexNet | GoogleNet | VGG16 | ResNet50 |
|---|---|---|---|---|---|---|
| Device | CPU | | GPU | | | |
| Training accuracy | 92.93 | 91.26 | 90.45 | 85.60 | 96.30 | 96.18 |
| Test accuracy | 74.01 | 73.96 | 76.55 | 23.00 | 64.21 | 57.95 |
| Difference | 18.92 | 17.30 | 13.90 | 62.60 | 32.09 | 38.23 |
| Training time | 157.83 | 78.26 | 870.68 | 1850.24 | 10255.02 | 4300.81 |

In this task, the training set consists of original images, while the testing set contains images with modified contrast. The significant drop in test accuracy compared to training accuracy suggests overfitting. GoogleNet, VGG16, and ResNet50 exhibited more overfitting compared to our model, while AlexNet demonstrated less overfitting. Therefore, both our model and AlexNet displayed greater robustness in handling contrast changes.

