# OpenReview forum: "FlyOrien: A bio-inspired model for incremental learning of object orientation"
_ICLR.cc/2025/Conference — Submitted to ICLR 2025_

### Official Review · Reviewer_kUzx · 2024-10-29

**Soundness:** 2
**Presentation:** 1
**Contribution:** 1
**Rating:** 3
**Confidence:** 4

**Summary:**

This paper proposes a bio-inspired model based on the mushroom body and central complex of the insect brain. Their model consists of a sparse coding layer followed by a continuous attractor network. They introduce two variations of a Hebbian learning rule for updating model parameters, allowing for fast training. The authors apply their model to the task of object orientation detection, and compare it to convolutional neural network baselines on the COIL-100 O and AS datasets, as well as a custom datasets collected from a quadruped robot.

**Strengths:**

- The bio-inspired method has local learning rules which enables fast training.

**Weaknesses:**

- I think there is a mismatch between the inspiration from biology (i.e. the insect brain) and the task setup (i.e. object orientation detection). I'm not convinced that insects/flies are evolved to estimate orientation of objects. Even if they would learn the orientation of objects, they'd typically only see a few orientations of an object, and then have to estimate a complete novel orientation; not first ingest a whole dataset of many orientations of a single object. A more intuitive application would be to predict the insect's own orientation / heading given the vision stream, which is effectively what bio-inspired navigation algorithms use a continuous attractor network for (e.g. https://ieeexplore.ieee.org/document/1307183, https://dl.acm.org/doi/10.1109/ICRA48506.2021.9560768 ). The final experiment goes in this direction, but still relies a 360 sweep first for "training" before it can start estimating its orientation.

- Given the experimental setup, the model basically needs to associate images to orientation bins. A different baseline that does this association (e.g. a Hopfield network) would be interesting to compare against.

- The train/test data is very correlated as the test data is basically gathered in the same "sweep" as the train data. Especially in the robot experiment it would be more convincing if the accuracy would be tested on other acquisitions with varying translation and distance of the robot to the object.

**Questions:**

- Looking at the examples of the COIL-100-AS dataset, it seems impossible to me to get a sensible orientation estimate given a symmetric object. Given that the methods still obtain a high accuracy for these objects seems to me that they are severely overfitting and looks rather undesired?

- For the robot experiments, how is the accuracy affected if the robot is put in a different position than the "train set"?

---

> ### Author Response · Authors · 2024-11-24
>
> Thank you very much for the insightful comments and suggestions. The paper is revised.
>
> ### Strengths: ###
>
>     •	The bio-inspired method has local learning rules which enables fast training.
>
> Thanks.
>
> ### Weaknesses: ###
>
>     •	I think there is a mismatch between the inspiration from biology (i.e. the insect brain) and the task setup (i.e. object orientation detection). I'm not convinced that insects/flies are evolved to estimate the orientation of objects. Even if they would learn the orientation of objects, they'd typically only see a few orientations of an object, and then have to estimate a complete novel orientation; not first ingest a whole dataset of many orientations of a single object. A more intuitive application would be to predict the insect's own orientation / heading given the vision stream, which is effectively what bio-inspired navigation algorithms use a continuous attractor network for (e.g. https://ieeexplore.ieee.org/document/1307183, https://dl.acm.org/doi/10.1109/ICRA48506.2021.9560768 ). The final experiment goes in this direction, but still relies a 360 sweep first for "training" before it can start estimating its orientation.
>
> Thank you for your insights. Our task does share some similarities with SLAM using CANN or place cells, as they all have bioinspired neural network components. However, our intention is not focused on the application of SLAM, but a more general and fundamental question, that is, if we can use neural mechanisms in the mushroom body to learn and identify orientation efficiently.
>
> A 360 sweep is not a strange strategy to collect views in insects. For example, the dung beetle dance is a behavior in which a dung beetle rotates on a dung ball to find direction while moving a dung ball, and an ant rotates near its nest before leaving the nest to remember views. For more details, please see:
>
> Dacke, M., Baird, E., El Jundi, B., Warrant, E. J., & Byrne, M. (2021). How Dung Beetles Steer Straight. Annual Review of Entomology, 66, 243–256. https://doi.org/10.1146/annurev-ento-042020-102149
>
> Wystrach, A., Philippides, A., Aurejac, A., Cheng, K., & Graham, P. (2014). Visual scanning behaviours and their role in the navigation of the Australian desert ant Melophorus bagoti. Journal of Comparative Physiology A: Neuroethology, Sensory, Neural, and Behavioral Physiology, 200(7), 615–626. https://doi.org/10.1007/s00359-014-0900-8
>
>
>     •	Given the experimental setup, the model basically needs to associate images to orientation bins. A different baseline that does this association (e.g. a Hopfield network) would be interesting to compare against.
>
> Thank you for the comment. Hopfield network, which is usually used as an example of associative learning, has a recurrent structure that forms attractors that attract states nearby a stable equilibrium to the equilibrium. Hence, its input and output are usually in the same space, while our task is not. So it is not straightforward to test the Hopfield network here.
>
>     •	The train/test data is very correlated as the test data is basically gathered in the same "sweep" as the train data. Especially in the robot experiment it would be more convincing if the accuracy would be tested on other acquisitions with varying translation and distance of the robot to the object.
>
> Thank you for the insightful comment. Yes, it will be more beneficial to test the performance of the model in a more complex setup. Due to the time limit, we are working on this towards the camera-ready version if accepted, or for our next publication.
>
> ### Questions: ###
>
>     •	Looking at the examples of the COIL-100-AS dataset, it seems impossible to me to get a sensible orientation estimate given a symmetric object. Given that the methods still obtain a high accuracy for these objects seems to me that they are severely overfitting and looks rather undesired?
>
> It is different from typical classification tasks. For typical classification tasks, images with the same label share similar features. However, in our dataset there is no correlation between the images with the same label, hence there is no way for generalization. With typical criteria, all of the models have to overfit to learn the dataset.
>
> Our model is functionally better because, as discussed in the paper, subtle features are important in learning samples in COIL-100-AS, however, CNNs shared weights to capture relationships between local features, which overshadow rare or unique patterns, thus harder to learn subtle features.
>
>     •	For the robot experiments, how is the accuracy affected if the robot is put in a different position than the "train set"?
>
> In the robot experiment that is with the simplified MB and without CANN, because the simplified MB cannot directly predict orientations that only appease in the testing set, the adjacent angle is used as the correct prediction criterion. Thus, Table 2 row 1 shows the accuracy if the robot is put in a different position.

---

> > ### Comment · Reviewer_kUzx · 2024-11-25
> >
> > Thank you for your clarification. I appreciate the references to motivate the 360 sweep. Looking at the dung beetle as well as desert ants, it strengthens me in the point that these insects rather use the MB to orient themselves, and/or to assess orientation of landmarks wrt themselves, instead of inferring the orientation of an object. This makes the link between the bio-inspiration and the experimental setup a bit far fetched.
> >
> > However, my main concern remains the setup of the train/test split, which are gathered in a single sweep as confirmed by the authors. I think this should be properly addressed to consider this for publication, and I retain my current score.

---

> ### Author Response · Authors · 2024-11-27
>
> Thank you for the new comments.
>
>     Thank you for your clarification. I appreciate the references to motivate the 360 sweep. Looking at the dung beetle as well as desert ants, it strengthens me in the point that these insects rather use the MB to orient themselves, and/or to assess orientation of landmarks wrt themselves, instead of inferring the orientation of an object. This makes the link between the bio-inspiration and the experimental setup a bit far fetched.
>
> There are similarities in setting up the tasks for using the MB to orient themselves, assessing the orientation of landmarks wrt themselves, and inferring the orientation of an object. Our model only infers the relative orientations between the observer and objects. Which class the task belongs to depends on whose absolute orientation is known.
>
> More specifically, assuming an observer always faces an object, with a reference direction which could be true north, there are three orientations: the angle the observer is facing $o$, the angle the object is facing $o^\prime$, and their relative angle $o - o^\prime$. Knowing any two allows the computation for the third. If $o$ and $o - o^\prime$ are known, it is an object-orienting problem; if $o^\prime$ and $o - o^\prime$ are known, it is an observer-orienting problem. For simplification, in our paper, in the object-orienting problem, $o$ is set to $0$, and in the observer-orienting problem, $o^\prime$ is set to $0$. Hence, in our dataset, there is only one number as a label for each sample, and the two problems need not be explicitly distinguished.
>
>     However, my main concern remains the setup of the train/test split, which are gathered in a single sweep as confirmed by the authors. I think this should be properly addressed to consider this for publication, and I retain my current score.
>
> The concern is reasonable in a typical classification task in which samples with the same samples are correlated. However,  in our dataset, there are images sharing the same label with various objects, which has no correlation between them. Hence, collecting data from a single weep does not make data correlated.
>
> It also does not hurt the fact that the baselines take much longer time and more epochs to learn on them, because the key difficulties, which are the low correlation and high interference between the samples with the same label, still exist with one sweep. If the data from a single sweep is easy, why baseline CNNs cannot do it efficiently? Right because the task is simple, the result more clearly suggests a fundamental ability missing from baseline CNNs: learning efficiently from subtle details.

---

> ### Author Response · Authors · 2024-12-03
>
> Based on your feedback, we revised our robot experiment. The robot now visits the same object four times, capturing images from four different distances. Images taken at 80 cm are used as the training set, while images from the other distances (70 cm, 90 cm, and 100 cm) form the testing set. The table below presents the results:
>
> |   Objects | Metric   | Distance | 70.0  | 80.0  | 90.0  | 100.0 |
> |--------------|----------|----------|-------|-------|-------|-------|
> | **cup**      | Top-2    |          | 23.6  | 80.6  | 61.1  | 54.2  |
> |              | Top-5    |          | 65.3  | 94.4  | 95.0  | 91.7  |
> | **foam**     | Top-2    |          | 45.8  | 87.5  | 63.9  | 41.7  |
> |              | Top-5    |          | 86.1  | 100.0 | 94.4  | 87.5  |
> | **plant**    | Top-2    |          | 30.6  | 90.3  | 37.5  | 48.6  |
> |              | Top-5    |          | 65.3  | 100.0 | 86.1  | 83.3  |
>
> These results indicate that our model can effectively predict the orientation during different visits. We hope this method of data separation addresses your concerns.

---

### Official Review · Reviewer_gqNV · 2024-10-31

**Soundness:** 2
**Presentation:** 3
**Contribution:** 2
**Rating:** 5
**Confidence:** 4

**Summary:**

This paper proposes a neural network model inspired by the neural circuits of insects, for incremental learning of object orientation. The model called FlyOrien, consists of two main parts mimicking the insect brain's Mushroom Body (MB) and Central Complex (CX). The MB part is responsible for sparse coding and associative learning, while the CX part integrates multiple inputs to refine the orientation detection. The model's performance is compared to typical CNN architectures. The model runs much faster than CNNs even when executed on CPU rather than GPU, and the its number of parameters is 250 times smaller. The performance is similar to CNNs and in some cases, better.
Simulations mainly include 2 datasets: a modified version of COIL-100 and a propriety dataset of real objects acquired by a quadruped robot.

**Strengths:**

The paper has a very good and comprehensive introduction, which includes a strong motivation for studying the suggested bio-inspired model, and an extended review of the reference biological mechanism of the insect's brain.
The paper also has a clear mathematical description of the model (simplified Mushroom Body) and a good explanation of its learning rules.
In the experimental section, the model is applied to a public dataset as well as a propriety dataset collected by a robot in a real environment.

**Weaknesses:**

The paper has several weaknesses, one of them is a long list of English typos, which may result from the use of an automatic language translation model. However, this is mainly annoying, and may be easily corrected.
Another weakness is the design of the simplified MB model, in particular the selection of the model size and parameters (e.g., number of KC units), that was adjusted to maximize the accuracy on the evaluation dataset. This actually shows a very high correlation between the model size (number of parameters) and the model accuracy, which may suggest that a much larger model may be needed to support larger images, a larger number of object categories and a realistic range of lighting conditions. It is not clear if the size of a more robust and general model will still be competitive and economic in comparison with SOTA CNNs. Also, the authors did not compare the model's performance to small CNN models such as SqueezeNet, MobileNet and others.
Another weakness is a poor description of the attractor network modeling the CX mechanism in section 2.2. Adding a visualization scheme to describe better the integration with the first part of the model (MB) may be helpful, as well as a better example than the one provided in figure 2.
Lastly, the authors include in the supplementary some analysis (figures A4-A6) on the resilience of the model to introducing additional new objects (overcoming the catastrophic forgetting problem of alternative models). However, the evaluation is based on datasets including only a small number of objects. An ablation study on the scalability of the model to cases with many more objects (hundreds) and more realistic scenes, are needed if one claims that the suggested model (or similar) should replace SOTA CNNs in robotic systems.

**Questions:**

As described in the Weaknesses section of the review, here are some specific questions and suggestions for the authors.
1. Provide a better ablation of the size and scalability of the model for robustness and realistic scenarios with larger number of objects and lighting conditions.
2. Compare the model's performance also to small CNN models such as SqueezeNet, MobileNet and possibly others, which has much less parameters than the currently evaluated CNNs (ResNet, AlexNet etc.).
3. Elaborate more on the modeling of the CX mechanism of the model (attractor network).
4. Correct the many typos in the paper (some examples: "sparse presentation", line 075; "inputs that are prepossessed", line 080; "an model", line 097; "a activity", line 103; "rules that enables", line 139; "there are dieerneces between", line 165; "layer is consists", line 171; "activities can active", line 201; etc.)
5. The equation in line 190 does not describe the mean of the samples, but merely the sum.
6. Lines 209-210 say: "Since a “KC” that is always active provides little useful information, we implemented a threshold to disable such “KCs”."
    It is not clear why you need this. Is this action taken because of replacing spiking neurons with firing-rate ones? If not, how is this issue resolved in the insect brain?
7. In equations (5) and (6) lines 244 and 254, the sentence "if j is the label and k actives," is not clear.
8. In figures A4 and A5 of the appendix, the y-axis title reads "training times", but the figures' caption says "the y-axis represents the index of the object being retrieved.". You need to clarify this discrepancy and modify either the titles or the captions.

---

> ### Author Response · Authors · 2024-11-24
>
> Thank you very much for the insightful comments and constructive suggestions. The paper is revised according to your comments and explained in the following response.
>
>
> ### Strengths: ###
>     The paper has a very good and comprehensive introduction, which includes a strong motivation for studying the suggested bio-inspired model, and an extended review of the reference biological mechanism of the insect's brain. The paper also has a clear mathematical description of the model (simplified Mushroom Body) and a good explanation of its learning rules. In the experimental section, the model is applied to a public dataset as well as a propriety dataset collected by a robot in a real environment.
>
> Thanks.
>
>  ### Weaknesses: ###
>
>     The paper has several weaknesses, one of them is a long list of English typos, which may result from the use of an automatic language translation model. However, this is mainly annoying, and may be easily corrected.
>
> Sorry for the typos. We have checked the whole paper to identify and revise the typos.
>
>     Another weakness is the design of the simplified MB model, in particular the selection of the model size and parameters (e.g., number of KC units), that was adjusted to maximize the accuracy on the evaluation dataset. This actually shows a very high correlation between the model size (number of parameters) and the model accuracy, which may suggest that a much larger model may be needed to support larger images, a larger number of object categories and a realistic range of lighting conditions. It is not clear if the size of a more robust and general model will still be competitive and economic in comparison with SOTA CNNs.
>
> The experiments presented in Figure 4 was to illustrate the reason why we chose 10240 neurons, with which the performance of our model converged but still has a smaller number of trainable parameters than the baseline CNNs including MobileNet and Shufflenet. With more neurons, the performance should be better, as there are more parameters to store learned information. However, as the performance has almost converged, the improvement will be minor.
>
> The original COIL has 7200 images, which is not a large number. We are still trying to find or create a larger dataset for a similar task. However, some earlier works investigating MB-inspired Local Sensitivity Hashing model on larger datasets suggested that the feature captured by the random connections from “PN” to “KC” form random feature detectors on images, and more “KCs”, more likely the feature detectors are redundant, and accuracy converged along with more “KCs”. Hence, the number of “KCs” needed should not be linear to the number of samples. For more details, please see:
>
> Dasgupta, S., Stevens, C. F., & Navlakha, S. (2017). A neural algorithm for a fundamental computing problem. Science, 358(6364), 793–796. https://doi.org/10.1126/science.aam9868
>
> Wei, T., & Webb, B. (2022). DevFly : Bio-inspired Development of Binary Connections for Locality Preserving Sparse Codes. In Thirty-sixth Conference on Neural Information Processing Systems.
>
>     Also, the authors did not compare the model's performance to small CNN models such as SqueezeNet, MobileNet and others.
>
> Revised. We have done more experiments on MobileNet and Shufflenet, which are smaller than some of our baselines but slight larger than ours. The parameter numbers of MobileNet and Shufflenet are 1591656 and 973920, respectively, and ours is 737280. We are still working on implementing SqueezeNet or transplanting the existing code of SqueezeNet for the task. In the retrieval task, MobileNet and Shufflenet have better accuracy (99.98%, 99.51%) than our methods (92.93%, 91.26%) on ordinary objects, but worse in axisymmetric objects (79.81% 83.55%), (97.65% 97.86%), and took 9 to 15 times longer time to train.
>
>     Another weakness is a poor description of the attractor network modeling the CX mechanism in section 2.2. Adding a visualization scheme to describe better the integration with the first part of the model (MB) may be helpful, as well as a better example than the one provided in figure 2.
>
> Revised, the description is updated and a new figure is plotted to replace Figure 2 for a better illusion of the CX mechanism

---

> ### Author Response · Authors · 2024-11-24
>
> continued:
>
>     Lastly, the authors include in the supplementary some analysis (figures A4-A6) on the resilience of the model to introducing additional new objects (overcoming the catastrophic forgetting problem of alternative models). However, the evaluation is based on datasets including only a small number of objects. An ablation study on the scalability of the model to cases with many more objects (hundreds) and more realistic scenes, are needed if one claims that the suggested model (or similar) should replace SOTA CNNs in robotic systems.
>
> Thank you for this insightful comment. The COIL dataset contains 100 different objects and we have tested all of them which also show a high accuracy. In fact, the samples in the dataset were not shuffled during the training of our model in every experiment we showed in the paper, so the accuracy shown in these experiments is the accuracy when incremental learning is finalized. Due to the limited space on a page to show a table, we only presented the first objects in figures A4 to A5. For Fig A6, we choose to present the first four objects along with incremental learning on the whole dataset, as they are most likely to be forgotten because the models learn other objects for a longer time after learning them.
>
> We don’t have an intention to replace SOTA CNNs in robotic systems, as we have not tested all tasks related to CNNs, but it is interesting for further exploration.
>
> ### Questions: ###
>
>     As described in the Weaknesses section of the review, here are some specific questions and suggestions for the authors.
>
>     1.	Provide a better ablation of the size and scalability of the model for robustness and realistic scenarios with larger number of objects and lighting conditions.
>
> Please see the earlier response starting from "The experiments presented in Figure 4 ..."
>
>     2.	Compare the model's performance also to small CNN models such as SqueezeNet, MobileNet and possibly others, which has much less parameters than the currently evaluated CNNs (ResNet, AlexNet etc.).
>
> Please see an earlier response. MobileNet and Shufflenet are tested and compared.
>
>     3.	Elaborate more on the modelling of the CX mechanism of the model (attractor network).
>
>  Please see the earlier reply. Section 2.2 and Figure 2 are now revised as you commented.
>
>     4.	Correct the many typos in the paper (some examples: "sparse presentation", line 075; "inputs that are prepossessed", line 080; "an model", line 097; "a activity", line 103; "rules that enables", line 139; "there are dieerneces between", line 165; "layer is consists", line 171; "activities can active", line 201; etc.)
>
> Thank you for pointing out the typos in the paper. We have carefully reviewed the manuscript and corrected the issues you identified and others.
>
>     5.	The equation in line 190 does not describe the mean of the samples, but merely the sum.
>
> Thanks. It is revised to be $\bar{x}=\sum_{i=0}^{d}{x_{i}}/d$.
>
>     6.	Lines 209-210 say: "Since a “KC” that is always active provides little useful information, we implemented a threshold to disable such “KCs”." It is not clear why you need this. Is this action taken because of replacing spiking neurons with firing-rate ones? If not, how is this issue resolved in the insect brain?
>
> This action is taken because our model ignores the neurodynamic of KCs for habituation or adaptation. This action is taken as a simple approximation of habituation, which is a form of non-associative learning by which neurons’ response to a repeated, benign stimulus decreases over time. For a brief review, please see:
>
> Groves, P. M., & Thompson, R. F. (1970). Habituation: A dual-process theory. Psychological Review, 77(5), 419–450. https://doi.org/10.1037/h0029810
>
>     7.	In equations (5) and (6) lines 244 and 254, the sentence "if j is the label and k actives," is not clear.
>
> It is revised in equations (5) and (6) and the nearby texts.
>
>     8.	In figures A4 and A5 of the appendix, the y-axis title reads "training times", but the figures' caption says "the y-axis represents the index of the object being retrieved.". You need to clarify this discrepancy and modify either the titles or the captions.
>
> Thanks for identifying the discrepancy. In the figure, the n-th row shows after training images of n objects, the retrieval accuracy on the 1-th to n-th objects. A better term is “trained objects”.  The figure is revised.

---

> > ### Author Response · Authors · 2024-11-27
> >
> > Thank you for reading our responses and updating the rating. Could you please provide a few more comments explaining the rating change so we can improve our paper further?

---

> > > ### Comment · Reviewer_gqNV · 2024-11-27
> > > **Overall rating change**
> > >
> > > Indeed, you have provided some answers and corrections following the questions and comments in my review. However, the concerns raised by reviewer kUzx, which are inline with my own concerns about the experimental setup, were not dismissed. The answers to these concerns about the experiments and dataset were not satisfactory. While I was convinced that the bio-inspired mechanism is indeed of interest and should be explored for AI and robotics, the current experimental setup is very limited and cannot validate the potential advantages of the MB mechanism of the insect brain, in comparison with existing deep net architectures.

---

> ### Author Response · Authors · 2024-11-27
>
> Thank you for clarifying it.
>
> We do not intend to claim that MB-inspired circuits are generally better than DNNs by this work. Instead, our goal is to reveal that the MB-like structure can efficiently learn datasets with low correlations between samples sharing the same label—datasets that typical CNNs struggle to learn efficiently, and the result supports it. Therefore, it is worthwhile to investigate the underlying mechanisms that enable efficient learning from subtle details and consider incorporating these mechanisms into ANNs. We can introduce more data samples the improve the correlation, however, it makes the task collapse back towards a typical classification task, and the difference between MB-inspired structure and ANNs be covered.

---

> ### Author Response · Authors · 2024-12-03
>
> Based on your feedback, we revised our robot experiment. The robot now visits the same object four times, capturing images from four different distances. Images taken at 80 cm are used as the training set, while images from the other distances (70 cm, 90 cm, and 100 cm) form the testing set. The table below presents the results:
>
> |   Objects | Metric   | Distance | 70.0  | 80.0  | 90.0  | 100.0 |
> |--------------|----------|----------|-------|-------|-------|-------|
> | **cup**      | Top-2    |          | 23.6  | 80.6  | 61.1  | 54.2  |
> |              | Top-5    |          | 65.3  | 94.4  | 95.0  | 91.7  |
> | **foam**     | Top-2    |          | 45.8  | 87.5  | 63.9  | 41.7  |
> |              | Top-5    |          | 86.1  | 100.0 | 94.4  | 87.5  |
> | **plant**    | Top-2    |          | 30.6  | 90.3  | 37.5  | 48.6  |
> |              | Top-5    |          | 65.3  | 100.0 | 86.1  | 83.3  |
>
> These results indicate that our model can effectively predict the orientation during different visits. We hope this method of data separation addresses your concerns.

---

### Official Review · Reviewer_yQ2x · 2024-11-04

**Soundness:** 2
**Presentation:** 2
**Contribution:** 2
**Rating:** 5
**Confidence:** 4

**Summary:**

This paper introduces FlyOrien, a bio-inspired model for object orientation detection that takes inspiration from insect neural circuits, specifically the Mushroom Body (MB) for sparse coding and the Central Complex (CX) for orientation integration. FlyOrien is designed for incremental learning, allowing it to recognize object orientations efficiently after minimal exposure while avoiding catastrophic forgetting. It was tested on a modified version of the COIL-100 dataset and real-world robotic tasks, showing strong performance in orientation retrieval and prediction, with reduced computational needs compared to CNNs.

**Strengths:**

The paper introduces FlyOrien, a bio-inspired model for learning object orientation, drawing from insect brain structures like the mushroom body and central complex. FlyOrien mimics biological neural circuits instead of traditional CNNs to learn object orientation after a single exposure. The model is well-explained, and experiments are robust, showing FlyOrien’s efficiency on modified datasets and real-world robotic tasks. Some sections could benefit from clearer explanations, particularly on specific model components.

The paper is mostly clear, though many structural and grammatical adjustments are needed to improve readability. FlyOrien’s lightweight, GPU-free design makes it a potential tool for resource-limited devices like drones, with promising applications in navigation, tracking, and surveillance.

In summary, FlyOrien offers an original approach, but a deeper comparison with existing methods and greater clarity in a few areas would strengthen the paper.

**Weaknesses:**

The paper could benefit from a more detailed comparison with existing orientation detection methods to better highlight its novelty and contributions. Claims about the model’s generalization to multimodal inputs (like olfactory and directional cues) are unsupported by the presented experiments. Some model components, like the modifications to the Continuous Attractor Neural Network, lack detail, making the implementation hard to fully understand or replicate. Also, the use of different hardware (CPU for FlyOrien and GPU for CNNs) creates an uneven comparison; training both on the same hardware would provide a clearer performance evaluation. Finally, grammatical and structural adjustments throughout would improve clarity and readability.


Clarity and Consistency
- Some phrases could be clarified. For example, use "visual orientation detection" instead of “orientation detection with vision.”
- The phrase "eliminating the need for large datasets and repeated training" might be too strong and could be replaced with "reducing the need for large datasets and repeated training."
- The statement “Our intention to retrieve orientation is related to object pose estimation, but different in many aspects” is vague and does not clearly delineate the differences.
- Claims like "perfect memory" are unrealistic and may misrepresent existing methods.
- The description "the same images are presented in the test" is confusing. Typically, retrieval involves querying with existing data, not reusing the same images. Please clarify whether the task involves retrieving orientations for previously seen images or something else.
- The term "catastrophic forgetting" is typically used in the context of incremental learning, not in relation to cross-validation. Please clarify the context in which catastrophic forgetting occurs or use a more appropriate term if applicable.

Methodology and Justification
- In the introduction, it would be beneficial to provide a short summary of the proposed method and how it differentiates from existing approaches.
- Briefly summarize the structure of the paper in the introduction to guide readers through the upcoming sections.
- The descriptions of Method 1 and Method 2 are vague and lack technical clarity.
- You assumed “MB uses similar ways for coding, and experiments shown it is functional.” The assumption is stated without sufficient justification or reference to supporting experiments.
- The modifications made to the Continuous Attractor Neural Network (CANN) for the CX component are not clearly explained.
- You mentioned “showing that it outperforms traditional artificial neural networks in terms of efficiency.” The term "efficiency" is broad and undefined. Specify which metrics are being referred to.
- The phrase “Both the first and second half of our model are set as a multi-class classification problem.” The distinction between the "first half" and "second half" is unclear.

Evaluation Criteria and Metrics
- "FlyOrien’s effectiveness in dataset and real-world applications" lacks specifics. The evaluation criteria, metrics used, and how effectiveness is measured are not clearly defined.
- Using the Top-5 accuracy metric for retrieval tasks is unconventional. Retrieval tasks often use metrics like precision, recall, or mean average precision. Please justify the use of Top-5 accuracy or consider using more standard retrieval metrics.
- You mentioned "FlyOrien is designed to learn object orientations efficiently after a single exposure and can generalize to multimodal inputs, such as visual, olfactory, and directional cues." However, the experiments described focus primarily on visual inputs, so claims about generalization to multimodal inputs are unsupported by the presented experiments.
- Comparing a model trained on a CPU to CNNs trained on a GPU is misleading because hardware significantly impacts training time and performance metrics.

Grammar and Typos
- There are several typos, including:
  1. “whose underline mechanism” should be “underlying mechanism.”
  2. “orientation but in the plan of the image” should be “plane of the image.”
  3. “dieerences” should be “differences.”
- Missing spaces are common throughout the paper.
- There are issues with Subject-Verb Agreement, verb tense inconsistencies, incorrect article usage, redundant words, and typos.
- Train on CPU not “trained in CPU.” Revising the sentence could improve clarity.
- Ensure clarity in phrases such as "a dataset modified for orientation detection." Specify the original dataset and the modifications made.

Structure and Flow
- The paper lacks a Conclusion section.
- Some sentences end abruptly without concluding the thought.
- The sections jump between high-level descriptions and detailed technical explanations without clear transitions. Add transitions to maintain logical flow.
- Conduct a thorough proofreading to correct typographical and grammatical errors.

**Questions:**

1. Could you clarify the claim about multimodal input generalization? The current evidence appears to be limited to visual data testing only.

2. The speed comparison between FlyOrien (CPU) and CNNs (GPU) needs clarification. How was the 45x speed difference measured considering the different hardware platforms used?

3. Please specify:
   - Whether the 45x speed improvement refers to training time, inference time, or both
   - How hardware differences were accounted for in this comparison

4. Can you elaborate on the specific modifications made to the CANN in the CX component? Please include details about how these changes contribute to orientation interpolation.

5. The claim of "perfect memory" requires substantiation. What evidence supports this characterization?

6. The description "the connection is almost random" needs clarification:
   - What metrics were used to determine this?
   - What evidence supports this characterization of the connections?

---

> ### Author Response · Authors · 2024-11-24
>
> Thank you very much for the insightful comments and constructive suggestions.
>
> We appreciate your feedback that our model is well-explained and our experiments robust, demonstrating FlyOrien's efficiency on modified datasets and real-world robotic tasks. Your recognition of its potential applications in resource-limited devices like drones is encouraging.
>
> We also thank you for pointing out the weaknesses, and we have revised our paper according to them or explain in the following response:
>
> ## Clarity and Consistency ##
>
>     •	Some phrases could be clarified. For example, use "visual orientation detection" instead of “orientation detection with vision.”
>
>     •	The phrase "eliminating the need for large datasets and repeated training" might be too strong and could be replaced with "reducing the need for large datasets and repeated training."
>
>     •	The statement “Our intention to retrieve orientation is related to object pose estimation, but different in many aspects” is vague and does not clearly delineate the differences.
>
> Response: Revised to point out that our model focuses on the relative direction of an animal to an object on the ground.
>
>     •	Claims like "perfect memory" are unrealistic and may misrepresent existing methods.
>
> Response: The term is removed to avoid misrepresentation.
>
>     •	The description "the same images are presented in the test" is confusing. Typically, retrieval involves querying with existing data, not reusing the same images. Please clarify whether the task involves retrieving orientations for previously seen images or something else.
>
> Response: Retrieval usually involves coding of the original data which facilitate the access the original data again. Here in our task, the original image is not the data to store, but the orientation of the object relative the to the camera. Hence, we used the term retrieval for the process to access the memory for the previously learned orientation. If not reusing the same images accounts for retrieval, then we guess you’re mentioning the case that the orientation of the object is the data to store. It also makes sense for a specific small object. While we consider the definition, we also thought about the case that in a nature environment, the landmark might not one object, but a combination of multiple complex object that is hard to infer their direction if an insect come to it from another direction, such as unstructured rocks or brushes. Hence, we finally decide to treat the orientation associated with the view be the information to store, although we frequently use the term “object’s orientation”, which might cause the confusion.
>
> We also conducted the experiment to use different images from a very near spot to test the model to see if the model can find the correct label. The results are presented in section 3.2 Table 3.
>
>     •	The term "catastrophic forgetting" is typically used in the context of incremental learning, not in relation to cross-validation. Please clarify the context in which catastrophic forgetting occurs or use a more appropriate term if applicable.
>
> Response: Thank you for pointing it out. The term "catastrophic forgetting" in line 313 is distractive. This term in 313 is removed, and this concept is now only used in explaining incremental learning.
>
> ### Methodology and Justification ###
>
>     •	In the introduction, it would be beneficial to provide a short summary of the proposed method and how it differentiates from existing approaches.
>
> Response: The final paragraph in the introduction is revised for the summary and difference.
>
>     •	Briefly summarize the structure of the paper in the introduction to guide readers through the upcoming sections.
>
> Response: One more paragraph in the introduction is add for the summary.
>
>     •	The descriptions of Method 1 and Method 2 are vague and lack technical clarity.
>
> Response: Revised in Section 2.1.3.
>
>     •	You assumed “MB uses similar ways for coding, and experiments shown it is functional.” The assumption is stated without sufficient justification or reference to supporting experiments.
>
> Response: Ardin et al., 2016; Dasgupta et al., 2017; Wei et al., 2022; and Zhu et al., 2020 can support the claim. They are added into the paper.
>
>     •	The modifications made to the Continuous Attractor Neural Network (CANN) for the CX component are not clearly explained.
>
> Response: Revised, so as the figure (Fig 2) to illustrate the connections.
>
>     •	You mentioned “showing that it outperforms traditional artificial neural networks in terms of efficiency.” The term "efficiency" is broad and undefined. Specify which metrics are being referred to.
>
> Response: It is revised by comparing different epochs that our model and baseline models need for a similar performance.

---

> > ### Author Response · Authors · 2024-11-24
> >
> > •	The phrase “Both the first and second half of our model are set as a multi-class classification problem.” The distinction between the "first half" and "second half" is unclear.
> >
> > Response: The terms "first half" and "second half" are not necessary here. It is revised as “The object orientation detection is set as a multi-class classification problem.”, which is more straightforward and clearer.
> >
> > ### Evaluation Criteria and Metrics ###
> >
> >     •	"FlyOrien’s effectiveness in dataset and real-world applications" lacks specifics. The evaluation criteria, metrics used, and how effectiveness is measured are not clearly defined.
> >
> > Response: Revised in line 152. “Experiments show that FlyOrien is more efficient than traditional artificial neural networks, as it only needs a single epoch training to achieve Top-5 accuracy comparable to CNNs that typically converge after 100 epochs.”
> >
> >     •	Using the Top-5 accuracy metric for retrieval tasks is unconventional. Retrieval tasks often use metrics like precision, recall, or mean average precision. Please justify the use of Top-5 accuracy or consider using more standard retrieval metrics.
> >
> > Response: With a query in a conventional retrieval task, there are many data samples returning, and more than one sample can be correct. In this case, precision, recall, or mean average precision is more important than Top-5 accuracy. However, in our experiment, there is only one class is correct for a sample, hence, whether the class is in the Top-5 class is more important.
> >
> >     •	You mentioned, "FlyOrien is designed to learn object orientations efficiently after a single exposure and can generalize to multimodal inputs, such as visual, olfactory, and directional cues." However, the experiments described focus primarily on visual inputs, so claims about generalization to multimodal inputs are unsupported by the presented experiments.
> >
> > Response: It is from the fact that the MB in insects can process multimodal inputs, such as visual, olfactory, and directional cues. Currently, we are working on this point for our next paper. To avoid overstatement given the experiments presented in this paper, the claim is modified.
> >
> > After revision: “FlyOrien is designed to learn object orientations efficiently after a single exposure, and because it mimics the sparse coding of MB, it has the potential to generalize to multimodal inputs, such as posture, olfactory, and directional cues, which will be investigated in further research.”
> >
> >     •	Comparing a model trained on a CPU to CNNs trained on a GPU is misleading because hardware significantly impacts training time and performance metrics.
> >
> > Response: We are working on training them on the same platform. Due to the time limit, we are working on this towards the camera-ready version if accepted, or for our next publication.
> >
> > ### Grammar and Typos ###
> >
> >     •	There are several typos, including:
> >     1.	“whose underline mechanism” should be “underlying mechanism.”
> >     2.	“orientation but in the plan of the image” should be “plane of the image.”
> >     3.	“dieerences” should be “differences.”
> >     •	Missing spaces are common throughout the paper.
> >     •	There are issues with Subject-Verb Agreement, verb tense inconsistencies, incorrect article usage, redundant words, and typos.
> >     •	Train on CPU not “trained in CPU.” Revising the sentence could improve clarity.
> >
> > Response: Thank you for your careful inspection. The paper is carefully inspected and revised to correct typos and grammatical errors.
> >
> >     •	Ensure clarity in phrases such as "a dataset modified for orientation detection." Specify the original dataset and the modifications made.
> >
> > Response:  Revised, now it reads as “We tested FlyOrien on a dataset containing images labelled with orientations, which introduce strong interferences because images of the same object have different labels.”
> >
> > ### Structure and Flow ###
> >
> >     •	The paper lacks a Conclusion section.
> >
> > Response:  The original discussion section was intended to be “Discussion and Conclusion”. It is revised.
> >
> >     •	Some sentences end abruptly without concluding the thought.
> >     •	The sections jump between high-level descriptions and detailed technical explanations without clear transitions. Add transitions to maintain logical flow.
> >     •	Conduct a thorough proofreading to correct typographical and grammatical errors.
> >
> >
> > Response: The paper is carefully inspected and revised to correct typos and grammatical errors as well as for a better flow.

---

> > > ### Author Response · Authors · 2024-11-24
> > >
> > > ## Questions: ##
> > >
> > >     1.	Could you clarify the claim about multimodal input generalization? The current evidence appears to be limited to visual data testing only.
> > >
> > > Response: As answered in an earlier question, it is from the fact that the MB in insects can process multimodal inputs, such as visual, olfactory, and directional cues. Currently, we are working on this point for our next paper. To avoid overstatement given the experiments presented in this paper, the claim in our paper is modified.
> > >
> > >     2.	The speed comparison between FlyOrien (CPU) and CNNs (GPU) needs clarification. How was the 45x speed difference measured considering the different hardware platforms used?
> > >
> > > Response: The 45x speed difference is not a fair measurement and we are running CNNs on CPU and FlyOrien on GPU for a fair comparison. However, it does not hurt the conclusion that our model learns much faster because if a CNN runs on a CPU, it is much slower. Hence, the unfairness is on FlyOrien's side. Even though our model runs on a slower device, it is still faster.
> > >
> > >     3.	Please specify:
> > >     o	Whether the 45x speed improvement refers to training time, inference time, or both
> > >     Response: It was comparing for the training time.
> > >     o	How hardware differences were accounted for in this comparison
> > >
> > > Response: It was an unfair comparison, but does not hurt the conclusion that our model is faster in training than other models we compared because only our model runs on CPU, which is slower.
> > >
> > >     4.	Can you elaborate on the specific modifications made to the CANN in the CX component? Please include details about how these changes contribute to orientation interpolation.
> > >
> > > Response: We revised section 2.2 and Fig 2 to make it clearer. In short. MBONs feed to neurons in CANN that represent corresponding orientations, but there are more neurons between these neurons for finer orientations, thus the ring attractor dynamics by lateral excitation and global inhibition can interpolate by activating the neurons for finer orientations.
> > >
> > >     5.	The claim of "perfect memory" requires substantiation. What evidence supports this characterization?
> > >
> > > Response: Perfect memory can refer to saving original reference data for comparison in later steps. In our paper, it refers to the 3D CAD model that is stored in its original format without coding in a neural network. The term "perfect memory" is also used in previous research about visual information processing by the MB (such as Ardin et al., 2016), referring to saving the original image for comparison in later steps. To avoid confusion, the term “perfect memory” is removed from our paper.
> > >
> > > Ardin, P., Peng, F., Mangan, M., Lagogiannis, K., & Webb, B. (2016). Using an Insect Mushroom Body Circuit to Encode Route Memory in Complex Natural Environments. PLoS Computational Biology, 12(2), 1–22. https://doi.org/10.1371/journal.pcbi.1004683
> > >
> > >     6.	The description "the connection is almost random" needs clarification:
> > >     o	What metrics were used to determine this?
> > >
> > > Response: The connections from PNs to KCs are sparse and strong, thus the distribution of the connections is usually formulated as a combination problem. The randomness is about the combination of PNs nerve to a KC.
> > >
> > >     o	What evidence supports this characterization of the connections?
> > >
> > > Response: Caron et al., 2013; Baltruschat et al., 2021; Hayashi et al., 2022
> > > Caron, S. J. C., Ruta, V., Abbott, L. F., & Axel, R. (2013). Random convergence of olfactory inputs in the Drosophila mushroom body. Nature, 497(7447), 113–117. https://doi.org/10.1038/nature12063
> > >
> > > Baltruschat, L., Prisco, L., Ranft, P., Lauritzen, J. S., Fiala, A., Bock, D. D., & Tavosanis, G. (2021). Circuit reorganization in the Drosophila mushroom body calyx accompanies memory consolidation. Cell Reports, 34(11). https://doi.org/10.1016/j.celrep.2021.108871
> > >
> > > Hayashi, T. T., MacKenzie, A. J., Ganguly, I., Ellis, K. E., Smihula, H. M., Jacob, M. S., … Caron, S. J. C. (2022). Mushroom body input connections form independently of sensory activity in Drosophila melanogaster. Current Biology, 32(18), 4000-4012.e5. https://doi.org/10.1016/j.cub.2022.07.055

---

### Meta-Review · Area_Chair_ng28 · 2024-12-19

**Metareview:**

This paper proposes a bio-inspired method for object orientation detection. It received 3 critical reviews from experts, which appreciated the general idea but raised several issues:

- Justification of the contribution,
- Experimental setup, difficulty of benchmarks, soundness of the evaluation methodology,
- missing comparisons,
- Clarity and consistency.

The discussions, first between reviewers and authors, and then among reviewers and including the AC, quickly made a critical weakness emerge, namely that the dataset was generated from a single sweep. While the authors attempted to argue that there are only weak temporal correlations and that therefore train and test splits were not correlated, this was judged to not be admissible. The AC concurs and judges that the paper is not yet ready.

**Additional Comments On Reviewer Discussion:**

The reviewers engaged with the authors

---

### Decision · Program_Chairs · 2025-01-22

Reject